# Uncovering the computational mechanisms underlying many-alternative choice

Armin W Thomas[1,2,3,4], Felix Molter[2,3,5], Ian Krajbich[6]*

[1]Technische Universität Berlin, Berlin, Germany; [2]Freie Universität Berlin, Berlin, Germany; [3]Center for Cognitive Neuroscience Berlin, Berlin, Germany; [4]Max Planck School of Cognition, Berlin, Germany; [5]WZB Berlin Social Science Center, Berlin, Germany; [6]The Ohio State University, Columbus, United States

**Abstract** How do we choose when confronted with many alternatives? There is surprisingly little decision modelling work with large choice sets, despite their prevalence in everyday life. Even further, there is an apparent disconnect between research in small choice sets, supporting a process of gaze-driven evidence accumulation, and research in larger choice sets, arguing for models of optimal choice, satisficing, and hybrids of the two. Here, we bridge this divide by developing and comparing different versions of these models in a many-alternative value-based choice experiment with 9, 16, 25, or 36 alternatives. We find that human choices are best explained by models incorporating an active effect of gaze on subjective value. A gaze-driven, probabilistic version of satisficing generally provides slightly better fits to choices and response times, while the gaze-driven evidence accumulation and comparison model provides the best overall account of the data when also considering the empirical relation between gaze allocation and choice.

## Introduction

In everyday life, we are constantly faced with value-based choice problems involving many possible alternatives. For instance, when choosing what movie to watch or what food to order off a menu, we must often search through a large number of alternatives. While much effort has been devoted to understanding the mechanisms underlying two-alternative forced choice (2AFC) in value-based decision-making (*Alós-Ferrer, 2018*; *Bhatia, 2013*; *Boorman et al., 2013*; *Clithero, 2018*; *De Martino et al., 2006*; *Hare et al., 2009*; *Hunt et al., 2018*; *Hutcherson et al., 2015*; *Krajbich et al., 2010*; *Mormann et al., 2010*; *Philiastides and Ratcliff, 2013*; *Polanía et al., 2019*; *Rodriguez et al., 2014*; *Webb, 2019*) and choices involving three to four alternatives (*Berkowitsch et al., 2014*; *Diederich, 2003*; *Gluth et al., 2018*; *Gluth et al., 2020*; *Krajbich and Rangel, 2011*; *Noguchi and Stewart, 2014*; *Roe et al., 2001*; *Towal et al., 2013*; *Trueblood et al., 2014*; *Usher and McClelland, 2004*), comparably little has been done to investigate many-alternative forced choices (MAFC, more than four alternatives) (*Ashby et al., 2016*; *Payne, 1976*; *Reutskaja et al., 2011*).

Prior work on 2AFC has indicated that simple value-based choices are made through a process of gaze-driven evidence accumulation and comparison, as captured by the attentional drift diffusion model (*Krajbich et al., 2010*; *Krajbich and Rangel, 2011*; *Smith and Krajbich, 2019*) and the gaze-weighted linear accumulator model (GLAM; *Thomas et al., 2019*). These models assume that noisy evidence in favour of each alternative is compared and accumulated over time. Once enough evidence is accumulated for one alternative relative to the others, that alternative is chosen. Importantly, gaze guides the accumulation process, with temporarily higher accumulation rates for looked-at alternatives. One result of this process is that longer gaze towards one alternative should generally increase the probability that it is chosen, in line with recent empirical findings (*Amasino et al.,*

*For correspondence:
krajbich@gmail.com

Competing interests: The authors declare that no competing interests exist.

**eLife digest** In our everyday lives, we often have to choose between many different options. When deciding what to order off a menu, for example, or what type of soda to buy in the supermarket, we have a range of possibilities to consider. So how do we decide what to go for?

Researchers believe we make such choices by assigning a subjective value to each of the available options. But we can do this in several different ways. We could look at every option in turn, and then choose the best one once we have considered them all. This is a so-called 'rational' decision-making approach. But we could also consider each of the options one at a time and stop as soon as we find one that is good enough. This strategy is known as 'satisficing'.

In both approaches, we use our eyes to gather information about the items available. Most scientists have assumed that merely looking at an item – such as a particular brand of soda – does not affect how we feel about that item. But studies in which animals or people choose between much smaller sets of objects – usually up to four – suggest otherwise. The results from these studies indicate that looking at an item makes that item more attractive to the observer, thereby increasing its subjective value.

Thomas et al. now show that gaze also plays an active role in the decision-making process when people are spoilt for choice. Healthy volunteers looked at pictures of up to 36 snack foods on a screen and were asked to select the one they would most like to eat. The researchers then recorded the volunteers' choices and response times, and used eye-tracking technology to follow the direction of their gaze. They then tested which of the various decision-making strategies could best account for all the behaviour.

The results showed that the volunteers' behaviour was best explained by computer models that assumed that looking at an item increases its subjective value. Moreover, the results confirmed that we do not examine all items and then choose the best one. But neither do we use a purely satisficing approach: the volunteers chose the last item they had looked at less than half the time. Instead, we make decisions by comparing individual items against one another, going back and forth between them. The longer we look at an item, the more attractive it becomes, and the more likely we are to choose it.

*2019*; *Armel et al., 2008*; *Cavanagh et al., 2014*; *Fisher, 2017*; *Folke et al., 2017*; *Gluth et al., 2018*; *Gluth et al., 2020*; *Konovalov and Krajbich, 2016*; *Pärnamets et al., 2015*; *Shimojo et al., 2003*; *Stewart et al., 2016*; *Vaidya and Fellows, 2015*). While this framework can in theory be extended to MAFC (*Gluth et al., 2020*; *Krajbich and Rangel, 2011*; *Thomas et al., 2019*; *Towal et al., 2013*), it is still unknown whether it can account for choices from truly large choice sets.

In contrast, past research in MAFC suggests that people may resort to a 'satisficing' strategy. Here, the idea is that people set a minimum threshold on what they are willing to accept and search through the alternatives until they find one that is above that threshold (*McCall, 1970*; *Simon, 1955*; *Simon, 1956*; *Simon, 1957*; *Simon, 1959*; *Schwartz et al., 2002*; *Stüttgen et al., 2012*). Satisficing has been observed in a variety of choice scenarios, including tasks with a large number of alternatives (*Caplin et al., 2011*; *Stüttgen et al., 2012*), patients with damage to the prefrontal cortex (*Fellows, 2006*), inferential decisions (*Gigerenzer and Goldstein, 1996*), survey questions (*Krosnick, 1991*), risky financial decisions (*Fellner et al., 2009*), and with increasing task complexity (*Payne, 1976*). Past work has also investigated MAFC under strict time limits (*Reutskaja et al., 2011*). There, the authors find that the best model is a probabilistic version of satisficing in which the time point when individuals stop their search and make a choice follows a probabilistic function of elapsed time and cached (i.e., highest-seen) item value (*Chow and Robbins, 1961*; *Rapoport and Tversky, 1966*; *Robbins et al., 1971*; *Simon, 1955*; *Simon, 1959*).

There is some empirical evidence that points towards a gaze-driven evidence accumulation and comparison process for MAFC. For instance, individuals look back and forth between alternatives as if comparing them (*Russo and Rosen, 1975*). Also, frequently looking at an item dramatically increases the probability of choosing that item (*Chandon et al., 2009*). Empirical evidence has

further indicated that individuals use a gaze-dependent evidence accumulation process when making choices from sets of up to eight alternatives (*Ashby et al., 2016*).

Here, we sought to study the mechanisms underlying MAFC, by developing and comparing different versions of these models on choice, response time (RT), liking rating, and gaze data from a choice task with sets of 9, 16, 25, and 36 snack foods. These models combine an either passive or active account of gaze in the decision process with three distinct accounts of the decision mechanism, namely probabilistic satisficing and two variants of evidence accumulation, which either perform relative comparisons between the alternatives or evaluate each alternative independently.

In terms of overall goodness-of-fit, we find that the models with active gaze consistently outperform their passive-gaze counterparts. That is, gaze does more than bring an alternative into the consideration set, it actively increases the subjective value of the attended alternative relative to the others. The probabilistic satisficing model (PSM) performs slightly better than the other models at capturing individuals' choices and RTs, while the relative accumulator model provides the best overall account of the data after considering the observed positive relation between gaze allocation and choice.

## Results

### Experiment design

In each of 200 choice trials, subjects (N = 49) chose which snack food they would like to eat at the end of the experiment, out of a set of either 9, 16, 25, or 36 alternatives (50 trials per set size condition; see *Figure 1* and Materials and methods). We recorded subjects' choices, RTs, and eye movements. After the choice task, subjects also rated each food on an integer scale from −3 (i. e., not at all) to 3 (i.e., very much) to indicate how much they would like to eat each item at the end of the experiment (for an overview of the liking rating distributions, see *Figure 1—figure supplements 1*, *2*). We use these liking ratings as a measure of item value.

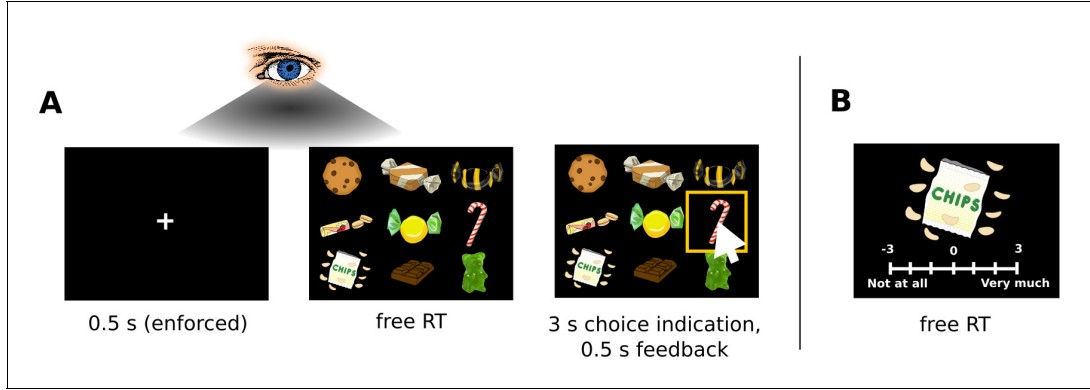

**Figure 1.** Choice task. (**A**) Subjects chose a snack food item (e.g., chocolate bars, chips, gummy bears) from choice sets with 9, 16, 25, or 36 items. There were no time restrictions during the choice phase. Subjects indicated when they had made a choice by pressing the spacebar of a keyboard in front of them. Subsequently, subjects had 3 s to indicate their choice by clicking on their chosen item with a mouse cursor that appeared at the centre of the screen. Subjects used the same hand to press the space bar and navigate the mouse cursor. For an overview of the choice indication times (defined as the time difference between the spacebar press and the click on an item), see *Figure 1—figure supplement 3*. Trials from the four set sizes were randomly intermixed. Before the beginning of each choice trial, subjects had to fixate a central fixation cross for 0.5 s. Eye movement data were only collected during the central fixation and choice phase. (**B**) After completing the choice task, subjects indicated how much they would like to eat each snack food item on a 7-point rating scale from −3 (not at all) to 3 (very much). For an overview of the liking rating distributions, see *Figure 1— figure supplements 1*, *2*. The tasks used real food items that were familiar to the subjects.

The online version of this article includes the following figure supplement(s) for figure 1:

**Figure supplement 1.** Liking rating distribution of each subject.

**Figure supplement 2.** Absolute (A–D) and relative (E–H; defined as the difference between an item's rating and the mean rating of the other items in a choice set) liking rating distributions for each set size.

**Figure supplement 3.** Choice indication times for each set size as indicated by the time difference between space bar press (indicating RT) and subsequent mouse click on a snack food item image.

## Visual search

To first establish a general understanding of the visual search process in MAFC, we performed an exploratory analysis of subjects' visual search behaviour (*Figures 2* and *3*). We define a gaze to an item as all consecutive fixations towards the item that happen without any interrupting fixation to other parts of the choice screen. Furthermore, we define the cumulative gaze of an item as the fraction of total trial time that the subject spent looking at the item (see Materials and methods).

All reported regression coefficients represent fixed effects from mixed-effects linear (for continuous dependent variables) and logistic (for binary dependent variables) regression models, which included random intercepts and slopes for each subject (unless noted otherwise). The 94% highest density intervals (HDI; 94% is the default in ArviZ 0.9.0 [*Kumar et al., 2019*] which we used for our analyses) of the fixed-effect coefficients are given in brackets, unless noted otherwise (see Materials and methods).

The probability that participants looked at an item in a choice set increased with the item's liking rating, while decreasing with set size (*Figure 2A–D*; $\beta = 2.0\%$, 94% HDI = [1.6, 2.3] per rating, $\beta = -1.4\%$, 94% HDI = [−1.5, −1.3] per item) (in line with recent empirical findings: *Cavanagh et al., 2019*; *Gluth et al., 2020*). Similarly, the probability that participants' gaze returned to an item increased with the item's rating while decreasing with set size (*Figure 2A–D*; $\beta = 1.6\%$, 94% HDI = [1.4, 1.8] per rating, $\beta = -0.65\%$, 94% HDI = [−0.74, –0.55] per item).

Gaze durations also increased with the item's rating (*Figure 2E–H*; $\beta = 11$ ms, 94% HDI = [8, 13] per rating) as well as over the course of a trial ($\beta = 0.79$ ms, 94% HDI = [0.36, 1.25] per additional gaze in a trial), while decreasing with set size ($\beta = -1.17$ ms, 94% HDI = [−1.39, –0.94] per item). Initial gazes to an item were generally shorter in duration than all later gazes to the same item in the same trial (*Figure 2I–L*; $\beta = -44$ ms, 94% HDI = [37, 51] difference between initial and returning gazes). Interestingly, the duration of the last gaze in a trial was dependent on whether it was to the chosen item or not (*Figure 2I–L*): last gaze durations to the chosen item were in general longer than last gaze durations to non-chosen items ($\beta = 162$ ms, 94% HDI = [122, 201] difference between last gazes to chosen and non-chosen items).

Next, we focused on subjects' visual search trajectories (*Figure 3*): For each trial, we first normalized time to a range from 0 to 100% and then binned it into 10% intervals. We then extracted the liking rating, position, and size for each item in a trial (see Materials and methods). An item's position was encoded by its column and row indices in the square grid (*Figure 1*; with indices increasing from left to right and top to bottom). All item attributes were centred with respect to their trial mean in the choice set (e.g., a centred row index of −1 in the set size with nine items represents the row one above the centre, whereas a centred item rating of −1 represents a rating one below the average of all item ratings in that choice set). For each normalized time bin, we computed a mixed-effects logistic regression model (see Materials and methods), regressing the probability that an item was looked at onto its attributes.

In general, subjects began their search at the centre of the screen (*Figure 3A,B*; as indicated by regression coefficients close to 0 for the items' row and column positions in the beginning of a trial), coinciding with the preceding fixation cross. Subjects then typically transitioned to the top left corner (*Figure 3A,B*; as indicated by increasingly negative regression coefficients for the items' row and column positions in the beginning of a trial) and then moved from top to bottom (*Figure 3B*; as indicated by the then increasingly positive regression coefficients for the items' row positions). Over the course of the trial, subjects generally focused their search more on highly rated (*Figure 3C*) and larger (*Figure 3D*) items, while the probability that their gaze returned to an item also steadily increased (*Figure 3E*; $\beta = 9.9\%$, 94% HDI = [9.2, 10.7] per second, $\beta = -0.73$, 94% HDI = [−0.80, –0.66] per item), as did the durations of these returning gazes (*Figure 3F*; $\beta = 14$ ms, 94% HDI = [12, 17] per second, $\beta = -2.9$ ms, 94% HDI = [−3.3, –2.5] per item). In general, the effects of item position and size on the search process decreased over time (*Figure 3A,B,D*). For exemplar visual search trajectories in each set size condition, see *Animations 1–4*.

Overall, the fraction of total trial time that subjects looked at an item was dependent on the liking rating, size, and position of the item, as well as the number of items contained in the choice set ($\beta = 0.5\%$, 94% HDI = [0.4, 0.6] per liking rating, $\beta = 0.02\%$, 94% HDI = [0.008, 0.03] per percentage increase in size, $\beta = -0.20\%$, 94% HDI = [−0.24, –0.15] per row position, $\beta = -0.044$, 94% HDI = [−0.075, –0.007] per column position, $\beta = -0.177$, 94% HDI = [−0.18, –0.174] per item).

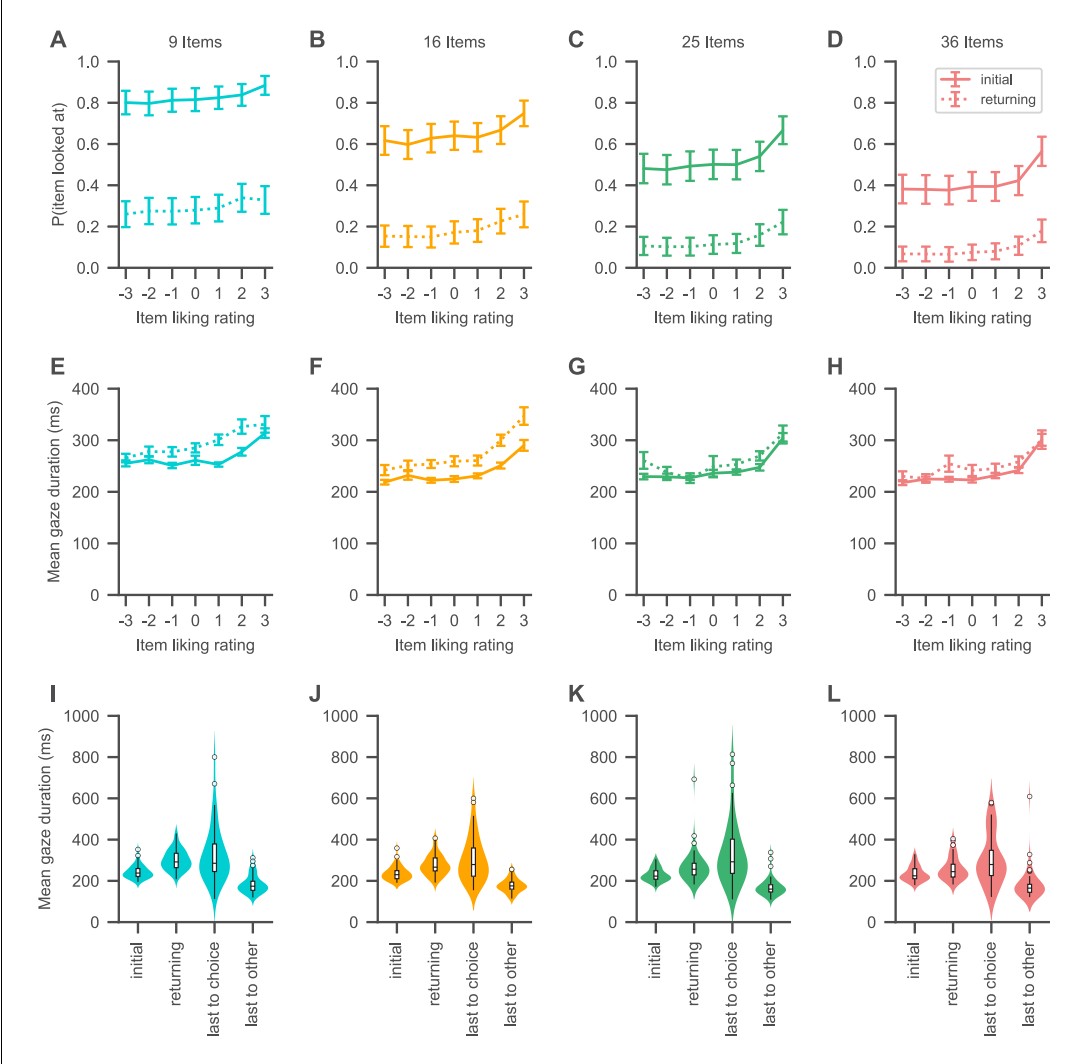

**Figure 2.** Gaze psychometrics for each set size. (A–H) The probability of looking at an item (A–D) as well as the mean duration of item gazes (E–H) increases with the liking rating of the item. Solid lines indicate initial gazes to an item, while dotted lines indicate all subsequent returning gazes to the item. (I–L) Initial gazes to an item are in general shorter in duration than all subsequent gazes to the same item in a trial. The last gaze of a trial is in general longer in duration if it is to the chosen item than when it is to any other item. See the Visual search section for the corresponding statistical analyses. Colours indicate set sizes. Violin plots show a kernel density estimate of the distribution of subject means with boxplots inside of them.

We also tested whether these item attributes influenced subjects' choice behaviour. However, the probability of choosing an item did not depend on the size or position of the item, but was solely dependent on the item's liking rating and the set size ($\beta$ = 3.9, 94% HDI = [3.5, 4.3] per liking rating, $\beta$ = 0.02, 94% HDI = [−0.015, 0.06] per percentage increase in item size, $\beta$ = −0.06, 94% HDI = [−0.12, 0.01] per row, $\beta$ = −0.03, 94% HDI = [−0.1, 0.03] per column, $\beta$ = −0.24, 94% HDI = [−0.25, –0.23] per item).

## Competing choice models

We consider the following set of decision models, spanning the space between rational choice and gaze-driven evidence accumulation.

The optimal choice model with zero search costs is based on the framework of rational decision-making (*Luce and Raiffa, 1957*; *Simon, 1955*). It assumes that individuals look at all the items of a choice set and then choose the best seen item with a fixed probability $\beta$, while making a probabilistic choice over the set of seen items (*J*) with probability *1-$\beta$* following a softmax choice rule ($\sigma$, with

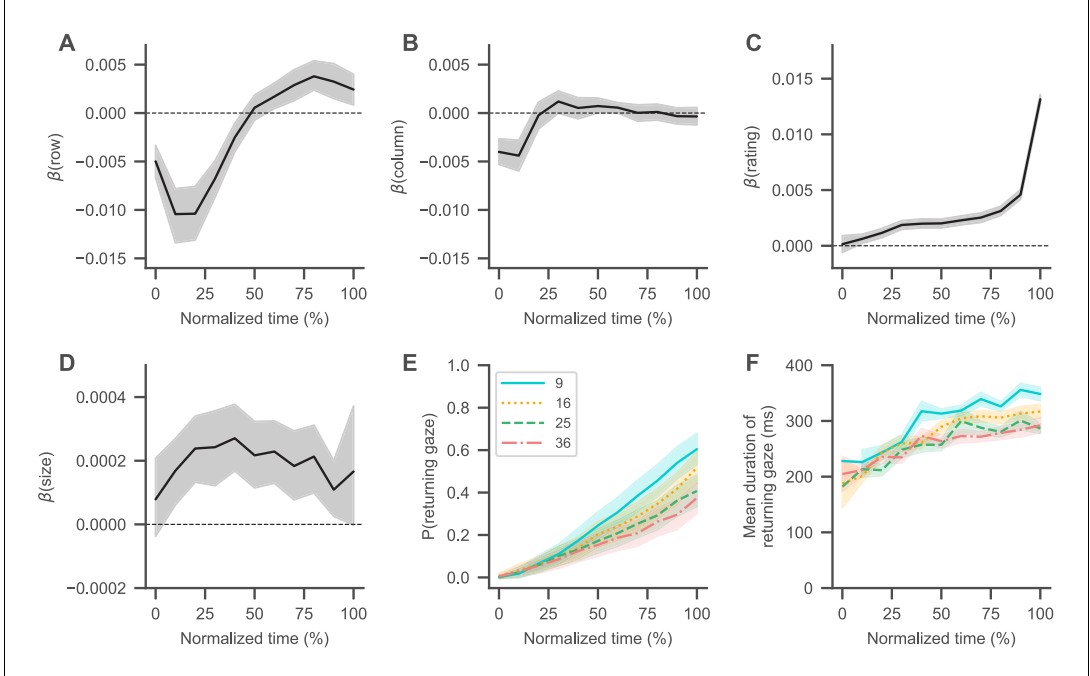

**Figure 3.** Visual search trajectory. (**A–D**) Black lines represent the fixed-effects coefficient estimates (with 94% HDI intervals surrounding them) of a mixed-effects logistic regression analysis (see Materials and methods) for each normalized trial time bin regressing the probability that an item was looked at onto its centred attributes (row (**A**) and column (**B**) position, liking rating (**C**), and size (**D**); see Materials and methods). Subjects generally started their search in the centre of the choice screen, coinciding with the fixation cross, and then transitioned to the top left corner (as indicated by decreasing regression coefficients for the items' row (**A**) and column positions (**B**)). From there, subjects generally searched from top to bottom (as indicated by slowly increasing regression coefficients for the items' row positions (**A**)), while also focusing more on items with a high liking rating (**C**) and a larger size (**D**). Dashed horizontal lines indicate a coefficient estimate of 0. (**E, F**) Over the course over a trial, subjects were more likely to look at items that they had already seen in the trial (**E**), while the duration of these returning gazes also increased (**F**). See the Visual search section for details on the corresponding statistical analyses. Coloured lines in (**E**) and (**F**) indicate mean values with standard errors surrounding them, while colours and line styles represent set size conditions.

inverse temperature parameter $\tau$) based on the items' values (*l*): $\sigma_i = \frac{exp(\tau \times l_i)}{\sum_{j \in J} exp(\tau \times l_j)}$.

The hard satisficing model assumes that individuals search until they either find an item with reservation value *V* or higher, or they have looked at all items (*Caplin et al., 2011*; *Fellows, 2006*; *McCall, 1970*; *Payne, 1976*; *Schwartz et al., 2002*; *Simon, 1955*; *Simon, 1956*; *Simon, 1957*; *Simon, 1959*; *Stüttgen et al., 2012*). In the former case, individuals immediately stop their search and choose the first item that meets the reservation value. Crucially, the reservation value can vary across individuals and set-size conditions. In the latter case, individuals make a probabilistic choice over the set of seen items, as in the optimal choice model.

Based on the findings by *Reutskaja et al., 2011*, we also considered a probabilistic version of satisficing, which combines elements from the optimal choice and hard satisficing models. Specifically, the PSM assumes that the probability with which individuals stop their search and make a choice at a given time point increases with elapsed time in the trial and the cached (i.e., highest-seen) item value. Once the search ends, individuals make a probabilistic choice over the set of seen items, as in the other two models (see Materials and methods).

Next, we considered an independent evidence accumulation model (IAM), in which evidence for an item begins accumulating once the item is looked at (*Smith and Vickers, 1988*). Importantly, each accumulator evolves independently from the others, based on the subjective value of the represented item. Once the accumulated evidence for an alternative reaches a predefined decision threshold, a choice is made for that alternative (much like deciding whether the item satisfies a reservation value) (see Materials and methods).

In line with many empirical findings (e.g., *Krajbich et al., 2010*; *Krajbich and Rangel, 2011*; *Lopez-Persem et al., 2016*; *Tavares et al., 2017*; *Smith and Krajbich, 2019*; *Thomas et al., 2019*),

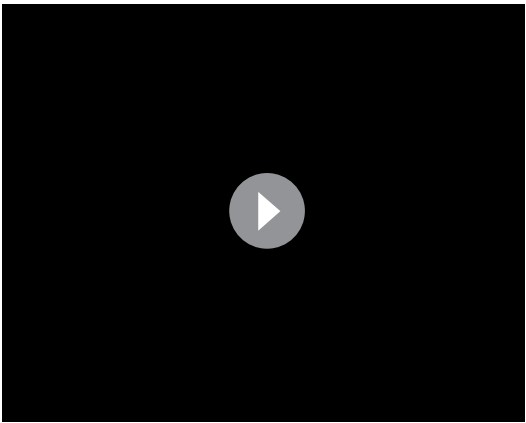

**Animation 1.** Exemplar visual search trajectory for choice sets with 9 alternatives. The video shows the visual search trajectory over the choice screen for one example trial . The current gaze position is indicated by a white box, while the choice is indicated by a red box. For better visibility, gaze durations have been increased by a factor of two.

https://elifesciences.org/articles/57012#video1

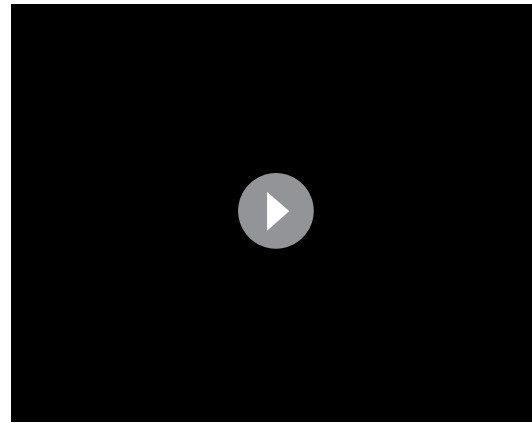

**Animation 2.** Exemplar visual search trajectory for choice sets with 16 alternatives. The video shows the visual search trajectory over the choice screen for one example trial. The current gaze position is indicated by a white box, while the choice is indicated by a red box. For better visibility, gaze durations have been increased by a factor of two.

https://elifesciences.org/articles/57012#video2

we also considered a relative evidence accumulation model (as captured by the GLAM; *Thomas et al., 2019*; *Molter et al., 2019*), which assumes that individuals accumulate and compare noisy evidence in favour of each item relative to the others. As with the IAM, a choice is made as soon as the accumulated relative evidence for an item reaches a predetermined decision threshold (see Materials and methods).

We further considered two different accounts of gaze in the decision process. The passive account of gaze assumes that gaze allocation solely determines the set of items that are being considered; an item is only considered once it is looked at. In contrast, the active account of gaze

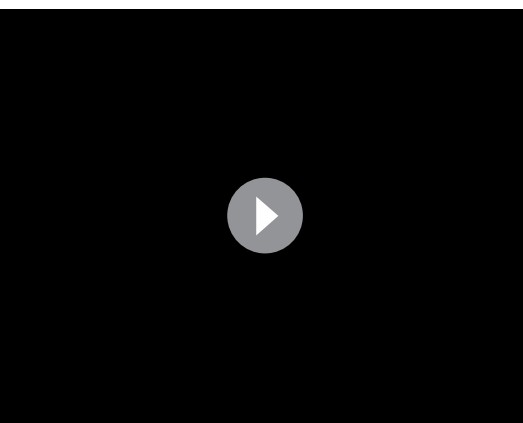

**Animation 3.** Exemplar visual search trajectory for choice sets with 25 alternatives. The video shows the visual search trajectory over the choice screen for one example trial. The current gaze position is indicated by a white box, while the choice is indicated by a red box. For better visibility, gaze durations have been increased by a factor of two.

https://elifesciences.org/articles/57012#video3

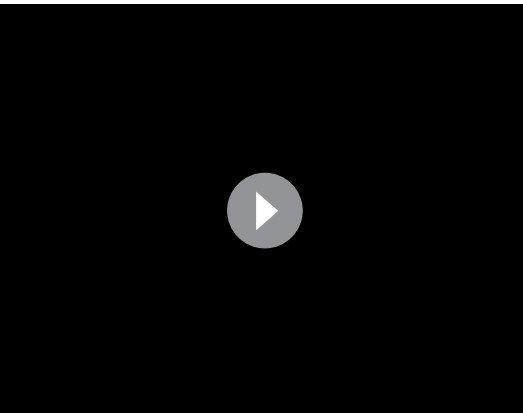

**Animation 4.** Exemplar visual search trajectory for choice sets with 36 alternatives. The video shows the visual search trajectory over the choice screen for one example trial. The current gaze position is indicated by a white box, while the choice is indicated by a red box. For better visibility, gaze durations have been increased by a factor of two.

https://elifesciences.org/articles/57012#video4

assumes that gaze influences the subjective value of an item in the decision process, thereby generating higher choice probabilities for items that are looked at longer. In the PSM, gaze time increases the subjective value of an item relative to the others. Similarly, in the accumulator models, the accumulation rate for an item (reflecting subjective value) increases relative to the others when the item is being looked at (for positively valued items).

Recent empirical findings indicate two distinct mechanisms through which gaze might actively influence these decision processes: multiplicative effects (*Krajbich et al., 2010*; *Krajbich and Rangel, 2011*; *Lopez-Persem et al., 2016* ; *Tavares et al., 2017*; *Smith and Krajbich, 2019*; *Thomas et al., 2019*) and additive effects (*Cavanagh et al., 2014*; *Westbrook et al., 2020*). Multiplicative effects discount the subjective values of unattended items (by multiplying them with $\gamma; 0 \leq \gamma \leq 1$), while additive effects add a constant boost ($\zeta; 0 \leq \zeta \leq 10$) to the subjective value of the attended item. Thus, multiplicative effects are proportional to the values of the items, while additive effects are constant for all items and independent of an item's value. We allow for both of these mechanisms in the modelling of the active influence of gaze on the decision process (see Materials and methods).

## Qualitative model comparison

First, we probed the assumptions of the optimal choice model with zero search costs, which predicts that subjects first look at all the items in a choice and then choose the highest-rated item at a fixed rate. Conditional on the set of looked-at items, subjects chose the highest-rated item at a very consistent rate across set sizes (*Figure 4A*; $\beta$ = 0.05%, 94% HDI = [−0.04, 0.14] per item), with an overall average of 84%. However, subjects did not look at all food items in a given trial (*Figure 4B*), while the fraction of items in a choice set that subjects looked at decreased across set sizes (*Figure 4B*; $\beta$ = −1.52%, 94% HDI = [−1.60, −1.45] per item) and their mean RTs increased (*Figure 4C*; $\beta$ = 85 ms, 94% HDI = [67, 102] per item). This immediately ruled out a strict interpretation of the optimal choice model, as subjects did not look at all items before making a choice.

Next, we tested the assumptions of the hard satisficing model, which predicts that subjects should stop their search and make a choice as soon as they find an item that meets their acceptance threshold. Accordingly, the last item that subjects look at should be the one that they choose (unless they look at every item). However, across set sizes, subjects only chose the last item that they looked at in 44.6% of the trials (*Figure 4D*; $\beta$ = 0.14%, 94% HDI = [−0.001, 0.26] per item). Even within the trials where subjects did not look at every item, the probability that they chose the last-seen item was on average only 44.1%.

The PSM, on the other hand, predicts that the probability with which subjects stop their search and make a choice increases with elapsed time and cached value (i.e., the highest-rated item seen so far in a trial). We found that both had positive effects on subjects' stopping probability, in addition to a negative effect of set size ($\beta$ = 2.7%, 94% HDI = [2.0, 3.3] per cached value, $\beta$ = 2.26%, 94% HDI = [1.69, 2.80] per second, $\beta$ = −0.22%, 94% HDI = [−0.24, −0.20] per item). Subjects' behaviour was therefore qualitatively in line with the basic assumptions of the PSM. Note that this finding does not allow us to discriminate between the PSM and evidence accumulation models, because both make very similar qualitative predictions about the relationship between stopping probability, time, and item value.

Last, we probed the behavioural association of gaze allocation and choice. To this end, we utilized a previously proposed measure of gaze influence (*Krajbich et al., 2010*; *Krajbich and Rangel, 2011*; *Thomas et al., 2019*): First, we regressed a choice variable for each item in a trial (1 if the item was chosen, 0 otherwise) on three predictor variables: the item's relative liking rating (the difference between the item's rating and the mean rating of all other items in that set) as well as the mean and range of the other items' liking ratings in that trial. We then subtracted this choice probability for each item in each trial from the empirically observed choice (1 if the item was chosen, 0 otherwise). This yields residual choice probabilities after accounting for the distribution of liking ratings in each trial. Finally, we computed the mean difference in these residual choice probabilities between items with positive and negative cumulative gaze advantages (defined as the difference between an item's cumulative gaze and the maximum cumulative gaze of any other item in that trial). This measure is a way to quantify the average increase in choice probability for the item that is looked at longest in each trial.

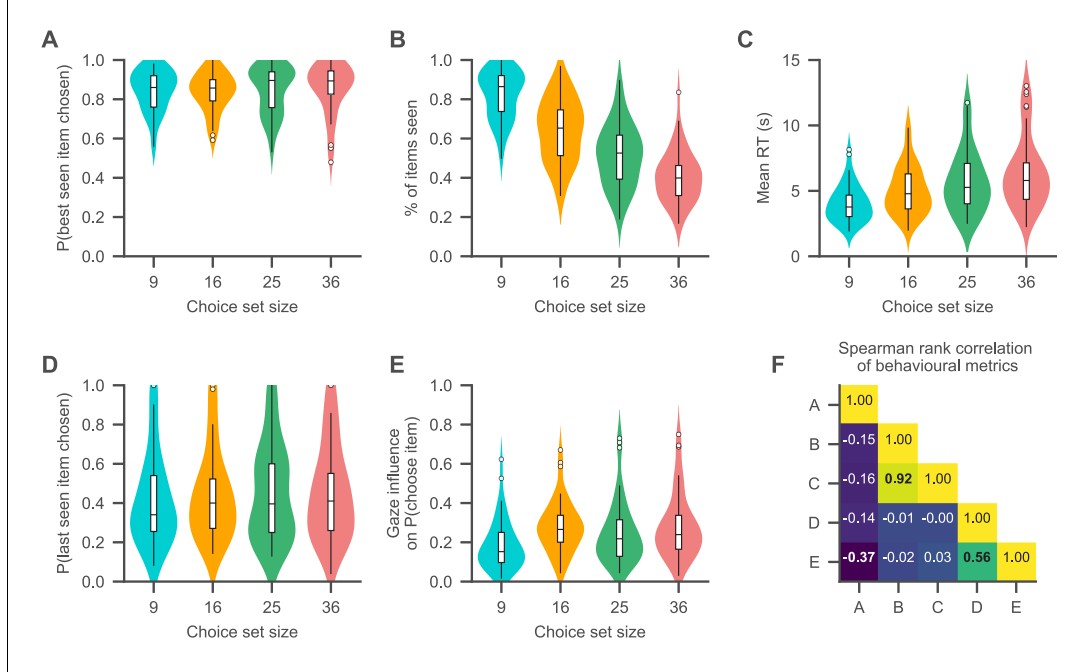

**Figure 4.** Choice psychometrics for each set size. (A) The subjects were very likely to choose one of the highest-rated (i.e., best) items that they looked in all set sizes. (B, C) The fraction of items of a choice set that subjects looked at in a trial decreased with set size (B), while subjects' mean RTs increased (C). (D) Subjects chose the item that they looked at last in a trial about half the time. (E) Subjects generally exhibited a positive association of gaze allocation and choice behaviour (as indicated by the gaze influence measure, describing the mean increase in choice probability for an item that is looked at longer than the others, after correcting for the influence of item value on choice probability; for details on this measure, see Qualitative model comparison). (F) Associations of the behavioural measures shown in (A–E) (as indicated by Spearman's rank correlation). Correlations are computed by the use of the pooled subject means across the set size conditions. Correlations with p-values smaller than 0.01 (Bonferroni corrected for multiple comparisons: 0.1/10) are printed in bold font. For a detailed overview of the associations of the behavioural measures, see *Figure 4—figure supplement 2*. See the Qualitative model comparison section for the corresponding statistical analyses. For a detailed overview of the associations between the behavioural choice measures and individuals' visual search, see *Figure 4—figure supplement 1*. Different colours in (A–E) represent the set size conditions. Violin plots show a kernel density estimate of the distribution of subject means with boxplots inside of them.

The online version of this article includes the following figure supplement(s) for figure 4:

**Figure supplement 1.** Association between the choice psychometrics presented in *Figure 4A–E* of the main text and a set of measures describing individuals' visual search behaviour.

**Figure supplement 2.** Detailed view of the associations of the choice psychometrics presented in *Figure 4F* of the main text.

We found that all subjects exhibited positive values on this measure in all set sizes (*Figure 4E*; with values ranging from 1.7% to 75%) and that it increased with set size (*Figure 4E*; β = 0.26%, 94% HDI = [0.15, 0.39] per item), indicating an overall positive association between gaze allocation and choice. In general, a subject's probability of choosing an item increased with the item's cumulative gaze advantage and the item's relative rating, while it decreased with the range of the ratings of the other items in a choice set and set size (β = 0.46%, 94% HDI = [0.4, 0.5] per percentage increase in cumulative gaze advantage, β = 3.6%, 94% HDI = [3.2, 4.0] per unit increase in relative rating, β = −2.8%, 94% HDI = [−3.1, –2.4] per unit increase in the range of ratings of the other items, β = −0.16, 94% HDI = [−0.18, −0.14] per item).

To further probe the assumption of gaze-driven evidence accumulation, we performed three additional tests: According to the framework of gaze-driven evidence accumulation, subjects with a stronger association of gaze and choice should generally also exhibit a lower probability of choosing the highest-rated item from a choice set (for a detailed discussion on this finding, see *Thomas et al., 2019*). For these subjects, the gaze bias mechanism can bias the decision process towards items that have a lower value but were looked at longer over the course of a trial. In line with this prediction, we found that probability of choosing the highest-rated seen item was negatively correlated with the gaze influence measure (β = −0.22%, 94% HDI = [−0.36, −0.08] per percentage increase in

gaze influence; the mixed-effects regression included a random slope and intercept for each set size).

Second, subjects with a stronger association between gaze and choice should be more likely to choose the last-seen item, as evidence for the looked-at item is generally accumulated at a higher rate. In line with this prediction, subjects with higher values on the gaze influence measure (indicating stronger gaze bias) were also more likely to choose the item that they looked at last in a trial ($\beta$ = 1.1%, 94% HDI = [0.9, 1.3] per percentage increase in gaze influence; the mixed-effects regression included a random slope and intercept for each set size).

Last, subjects with a stronger association of gaze and choice should be more likely to choose an item when it receives longer individual gazes. In line with previous work (e.g., *Krajbich et al., 2010*; *Krajbich and Rangel, 2011*), we investigated this by studying the probability of choosing the first-seen item in a trial as a function of the duration of the first gaze in that trial. Overall, this relationship was positive (as was the influence of the item's rating on choice probability), while the item's choice probability decreased with set size ($\beta$ = 18%, 94% HDI = [14, 22] per second, $\beta$ = 6.0%, 94% HDI = [5.5, 6.6] per rating, $\beta$ = −0.27%, 94% HDI = [−0.32, −0.22] per item).

## Relation of visual search and choice behaviour

To better understand the relation between visual search and choice behaviour, we also studied the association of the influence of an item's size, rating, and position on gaze allocation with the metrics of choice behaviour reported in *Figure 4* (namely, mean RT, fraction of looked-at items, probability of choosing the highest-rated seen item, and gaze influence on choice) (*Figure 4—figure supplement 1*). To quantify the influence of the item attributes on gaze allocation, we ran a regression for each subject of cumulative gaze (defined as the fraction of trial time that the subject looked at an item; scaled 0–100%) onto the four item attributes (row, column, size, and rating) and set size, resulting in one coefficient estimate ($\beta_{gaze}$) for the influence of each of the item attributes and set size on cumulative gaze.

Subjects with a stronger influence of rating on gaze allocation generally looked at fewer items ($\beta$ = −17%, 94% HPI = [−30, −5] per unit increase in $\beta_{gaze}$(rating); *Figure 4—figure supplement 1H*), were more likely to choose the highest-rated seen item ($\beta$ = 14%, 94% HDI = [5, 23] per unit increase in $\beta_{gaze}$(rating); *Figure 4—figure supplement 1P*), and were more likely to choose the last-seen item ($\beta$ = 40%, 94% HDI = [19, 61] per unit increase in $\beta_{gaze}$(rating); *Figure 4—figure supplement 1T*). Subjects with a stronger influence of item size on gaze allocation generally looked at fewer items ($\beta$ = −113%, 94% HDI = [−210, −6] per unit increase in $\beta_{gaze}$(size); *Figure 4—figure supplement 1G*), exhibited shorter RTs ($\beta$ = −18 s, 94% HDI = [−32, −4] per unit increase in $\beta_{gaze}$(size); *Figure 4—figure supplement 1K*), and were less likely to choose the last-seen item ($\beta$ = −189%, 94% HDI = [−377, −2] per unit increase in $\beta_{gaze}$(size); *Figure 4—figure supplement 1S*). Lastly, subjects with a stronger influence of column number (horizontal location) on gaze allocation generally exhibited longer RTs ($\beta$ = 3.96 s, 94% HDI = [0.40, 7.83] per unit increase in $\beta_{gaze}$(column); *Figure 4—figure supplement 1J*). This last effect mainly results from two behavioural patterns in the data: Subjects generally began their visual search in the upper-left corner of the screen (*Figure 3A,B*), resulting in more gaze for items that were located on the left side of the screen (i.e., with low column numbers). Thus, subjects who quickly decided also generally looked more at items on the left (resulting in a negative influence of column number on cumulative gaze). In contrast, slower subjects generally displayed more balanced looking patterns (resulting in no influence of column number on cumulative gaze). Taken together, these effects produce a positive association of mean RT and the influence of column number on cumulative gaze. We did not find any other statistically meaningful associations between visual search and choice metrics (*Figure 4—figure supplement 1*).

## Quantitative model comparison

Taken together, our findings have shown that subjects' choice behaviour in MAFC does not match the assumptions of optimal choice or hard satisficing, while it qualitatively matches the assumptions of probabilistic satisficing and gaze-driven evidence accumulation. To further discriminate between the evidence accumulation and probabilistic satisficing models, we fitted them to each subject's choice and RT data for each set size (see Materials and methods; for an overview of the parameter estimates, see *Figure 5—figure supplement 1* and *Supplementary files 1–3*) and compared their

fit by means of the widely applicable information criterion (WAIC; *Vehtari et al., 2017*). Importantly, we tested two variants of each of these models, one with a passive account of gaze in which gaze allocation solely determines the set of items that are considered in the decision process, and the other with an active account of gaze in which gaze affects the subjective value of the alternatives. In the active-gaze models (as indicated by the addition of a '+' to the model name), we allowed for both multiplicative and additive effects of gaze on the decision process (see Materials and methods). The model variants with a passive and active account of gaze were identical, other than for these two influences of gaze on subjective value. Note that all three model types can be recovered to a satisfying degree in our data (*Figure 5—figure supplement 2*).

According to the WAIC, the choices and RTs of the vast majority of subjects were best captured by the model variants with an active account of gaze (82% [40/49], 94% [46/49], 90% [44/49], and 86% [42/49] for 9, 16, 25, and 36 items respectively; *Figure 5A–D*). Specifically, the PSM+ won the plurality of individual WAIC comparisons in each set size (39% [19/49], 65% [32/49], 47% [23/49], and 51% [25/49] in the sets with 9, 16, 25, and 36 items, respectively), while the plurality of the remaining

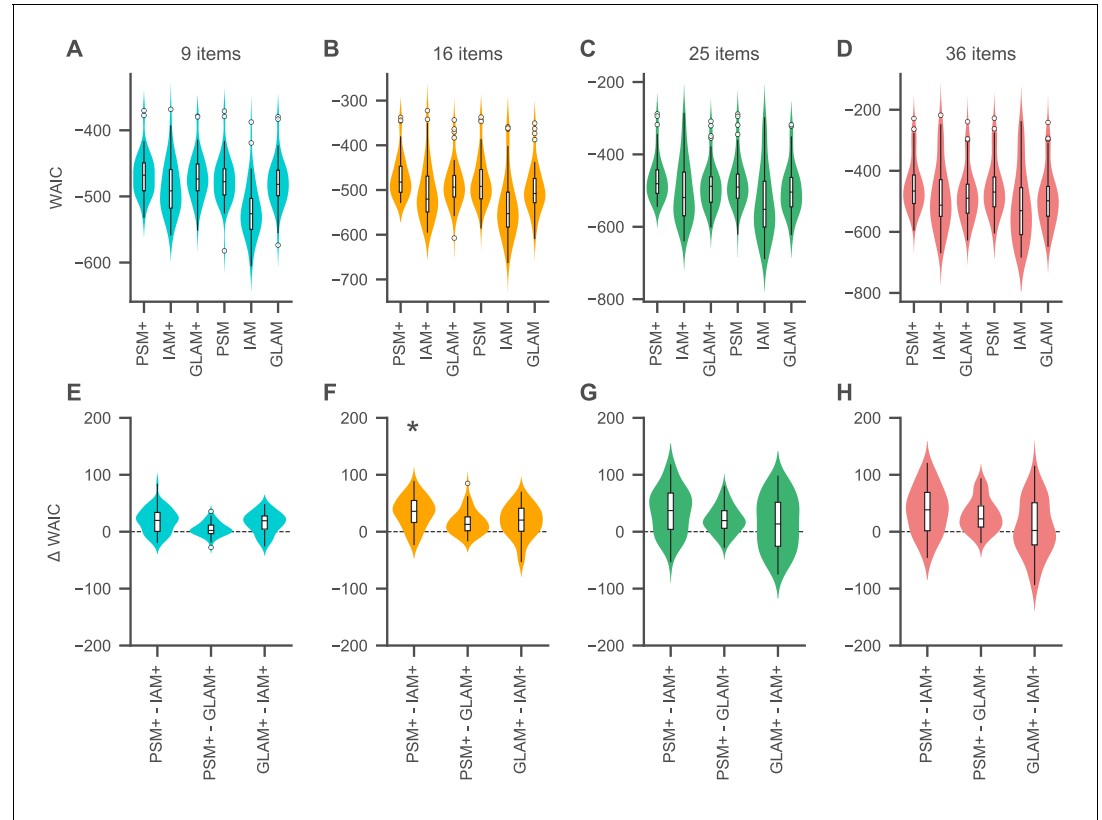

**Figure 5.** Relative model fit. (**A-D**) Individual WAIC values for the probabilistic satisficing model (PSM), independent evidence accumulation model (IAM), and gaze-weighted linear accumulator model (GLAM) for each set size. Model variants with an active influence of gaze are marked with an additional '+'. The WAIC is based on the log-score of the expected pointwise predictive density such that larger values in WAIC indicate better model fit. Violin plots show a kernel density estimate of the distribution of individual values with boxplots inside of them. (**E–H**) Difference in individual WAIC values for each pair of the active-gaze model variants. Asterisks indicate that the two distributions of WAIC values are meaningfully different in a Mann–Whitney U test with a Bonferroni adjusted alpha level of 0.0042 per test (0.05/12). See the Quantitative model comparison section for the corresponding statistical analyses. For an overview of the distribution of individual model parameter estimates, see *Figure 5—figure supplement 1* and *Supplementary files 1–3*. For an overview of the results of a model recovery of the three model types, see *Figure 5—figure supplement 2*. Colours indicate set sizes.

The online version of this article includes the following figure supplement(s) for figure 5:

**Figure supplement 1.** Parameter estimates from the in-sample fits of the probabilistic satisficing model (PSM; active-gaze variant: **A–E**, passive-gaze variant: **F–H**), GLAM (active-gaze variant: **I–M**, passive-gaze variant: **N–P**), and independent evidence accumulation model (IAM; active-gaze variant: **Q–T**, passive-gaze variant: **U**, **V**).

**Figure supplement 2.** Model recovery.

WAIC comparisons was won by the GLAM+ for 9 and 16 items (29% [14/49] and 16% [8/49] subjects, respectively) and by the IAM+ for 25 and 36 items (22% [11/49] and 24% [12/49] subjects, respectively).

To further test whether there was a winning model for each set size, we performed a comparison of the distributions of individual WAIC values resulting from each of the three active-gaze model variants. We used two-sided Mann–Whitney U tests with a Bonferroni adjusted alpha level of 0.0042 per test (0.05/12) (*Figure 5E–H*). This analysis revealed that the WAIC distributions of the PSM+ and GLAM+ were not meaningfully different from one another in any set size (U = 1287, p = 0.54; U = 1444, p=0.08; U = 1460, p = 0.07; U = 1469, p = 0.06 for 9, 16, 25, and 36 items), while the PSM+ was meaningfully better than the IAM+ for 16 items (U = 1705, p = 0.0003), but not for 9 (U = 1592, p = 0.005), 25 (U = 1573, p = 0.008), or 36 (U = 1508, p = 0.029) items. The GLAM+ was not meaningfully better than the IAM+ in any set size (U = 1518, p = 0.02; U = 1507, p = 0.03; U = 1388, p = 0.18; U = 1289, p = 0.53 for 9, 16, 25, and 36 items).

More generally, the difference in WAIC between the PSM+ and IAM+ as well as the PSM+ and GLAM+ increased with set size (β = 0.55, 94% HDI = [0.16, 0.93] per item for the IAM+ and β = 0.86, 94% HDI = [0.62, 1.12] per item for the GLAM+), indicating that the PSM+ provides an increasingly better fit to the choice and RT data as the set size increases. Notably, the corresponding fixed-effects intercept estimate was larger than 0 for the IAM+ (20, 94% HDI = [13, 27]), but not for the GLAM+ (−0.7, 94% HDI = [−5.3, 3.9]), suggesting that the PSM+ provides a meaningfully better fit for choice behaviour in smaller sets than the IAM+, but not than the GLAM+. Similarly, the WAIC-difference between the GLAM+ and IAM+ did not increase with set size (β = −0.30, 94% HDI = [−0.73, 0.15] per item), while the fixed-effects intercept estimate was meaningfully greater than 0 (21, 94% HDI = [15, 26]), indicating that the GLAM+ provides a meaningfully better fit than the IAM + for smaller sets.

Yet, WAIC only tells us about relative model fit. To determine how well each model fit the data in an absolute sense, we simulated data for each individual with each model and regressed the simulated mean RTs, probability of choosing the highest-rated item, and gaze influence on choice probability onto the observed subject values for each of these measures, in a linear mixed-effects regression analysis with one random intercept and slope for each set size (*Figure 6*; see Materials and methods). If a model captures the data well, the resulting fixed-effects regression line should have an intercept of 0 and a slope of 1 (as indicated by the black diagonal lines in *Figure 6*).

The PSM+ and GLAM+ both accurately recovered mean RT (*Figure 6A,C*; intercept = −138 ms, 94% HDI = [−414, 119], β = 1.01 ms, 94% HDI = [0.95, 1.05] per ms increase in observed RT for the PSM+; intercept = −51 ms, 94% HDI = [−327, 207], β = 0.97 ms, 94% HDI = [0.91, 1.06] per ms increase in observed RT for the GLAM+), while the IAM+ underestimated short and overestimated long mean RTs (*Figure 6B*; intercept = −1185 ms, 94% HDI = [−2293, −61], β = 1.29 ms, 94% HDI = [1.19, 1.39] per ms increase in observed RT).

All three models generally underestimated high probabilities of choosing the highest-rated item from a choice set (*Figure 6D–F*), while the PSM+ provided the overall most accurate account of this metric (*Figure 6D*; intercept = −1.90%, 94% HDI = [−7.64, 3.19], β = 0.85%, 94% HDI = [0.79, 0.91] per percentage increase in observed probability of choosing the highest-rated item), followed by the GLAM+ (*Figure 6E*; intercept = 11.66%, 94% HDI = [5.27, 17.41], β = 0.71%, 94% HDI = [0.64, 0.78] per percentage increase in observed probability of choosing the highest-rated item), and IAM + (*Figure 6F*; intercept = 10.48%, 94% HDI = [0.15, 28.20], β = 0.33%, 94% HDI = [0.15, 0.49] per percentage increase in observed probability of choosing the highest-rated item).

Turning to the gaze data, the PSM+ and IAM+ both slightly overestimated weak associations between gaze and choice while clearly underestimating stronger associations between them (*Figure 6G–H*; intercept = 7.03%, 94% HDI = [4.95, 10.23], β = 0.48%, 94% HDI = [0.38, 0.56] per percentage increase in observed gaze influence for the PSM+; intercept = 6.58%, 94% HDI = [4.28, 10.30], β = 0.38%, 94% HDI = [0.15, 0.49] per percentage increase in observed gaze influence for the IAM+). The GLAM+, in contrast, only slightly underestimated strong associations of gaze and choice (*Figure 6I*; intercept = −0.61%, 94% HDI = [−2.84, 1.64], β = 0.85%, 94% HDI = [0.77, 0.93] per percentage increase in observed gaze influence).

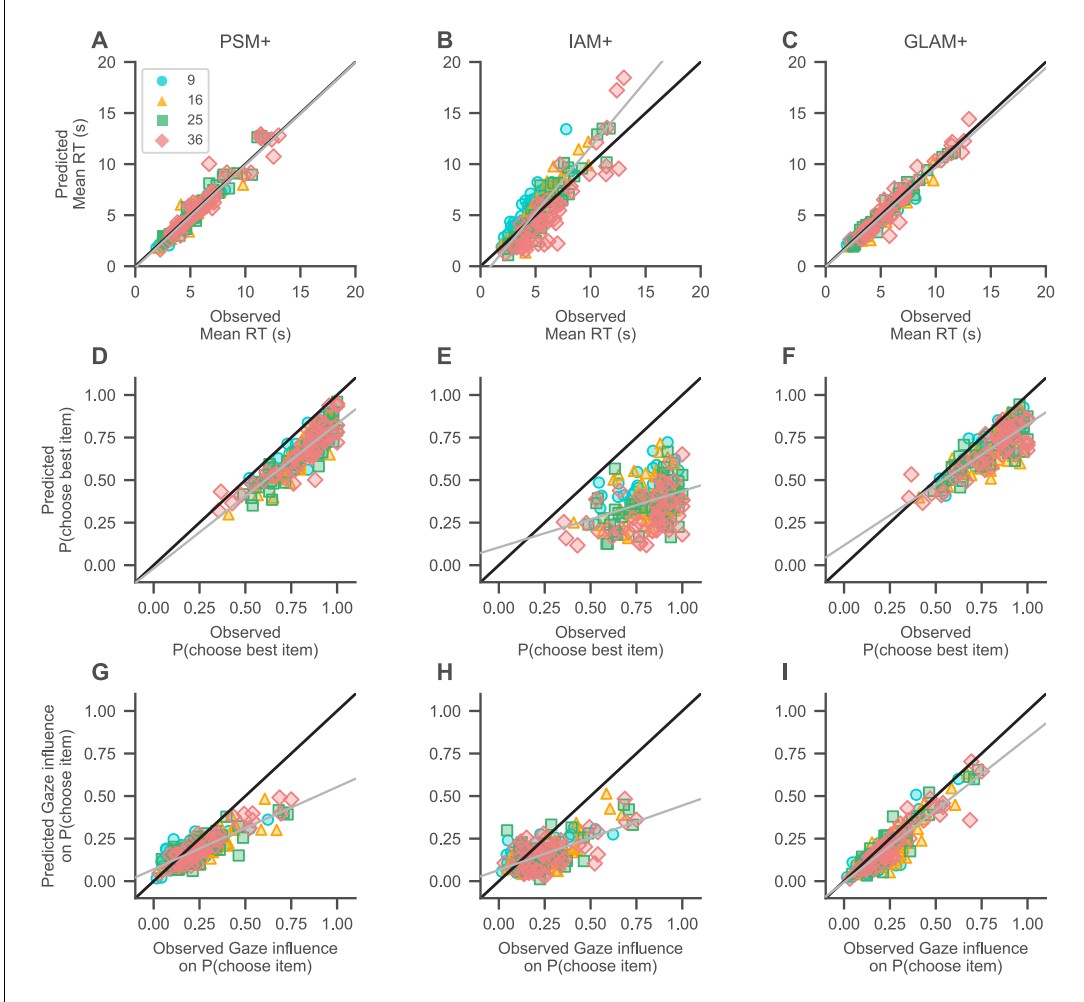

**Figure 6.** Absolute model fit. Predictions of mean RT (**A–C**), probability of choosing the highest-rated (i.e., best) item (**D–F**), and gaze influence on choice probability (**G–I**; for details on this measure, see Qualitative model comparison) by the active-gaze variants of the probabilistic satisficing model (PSM+; **A**, **G**, **D**), independent evidence accumulation model (IAM+; **B**, **E**, **H**), and gaze-weighted linear accumulator model (GLAM+; **C**, **F**, **I**). (**A–C**) The PSM+ and GLAM+ accurately recover mean RT, while the IAM+ underestimates short and overestimates long mean RTs. (**D–F**) The PSM+ provides the overall best account of choice accuracy, followed by the GLAM+, and IAM+. (**G–I**) The PSM+ and IAM+ clearly underestimate strong influences of gaze on choice; the GLAM+ provides the best account of this association and only slightly underestimates strong influences of gaze on choice. Gray lines indicate mixed-effects regression fits of the model predictions (including a random intercept and slope for each set size) and black diagonal lines represent ideal model fit. Model predictions are simulated using parameter estimates obtained from individual model fits (for details on the fitting and simulation procedures, see Materials and methods). See the Quantitative model comparison section for the corresponding statistical analyses. Colours and shapes represent different set sizes, while scatters indicate individual subjects.

## Gaze bias mechanism

To better understand the gaze bias mechanism in our data, we studied the gaze bias parameter estimates resulting from the individual model fits. Our modelling of the decision process included two types of gaze biases (see Materials and methods): multiplicative gaze biases (indicated by γ) proportionally discount the value of momentarily unfixated items (smaller γ values thus indicate stronger bias), while additive gaze biases (indicated by ζ) provide a constant increase to the value of the currently fixated item (larger ζ values thus indicate stronger bias).

We found evidence for both of these gaze biases in the PSM+ and GLAM+ across all set sizes, with mean (s.d.) γ estimates (per set size) of 9: 0.63 (0.29), 16: 0.53 (0.27), 25: 0.57 (0.27), and 36: 0.59 (0.29) for the PSM+ (*Supplementary file 1*) and 9: 0.72 (0.28), 16: 0.64 (0.27), 25: 0.68 (0.27), and 36: 0.71 (0.27) for the GLAM+ (*Supplementary file 3*). Similarly, mean (s.d.) ζ estimates were 9: 1.29 (1.62), 16: 1.73 (1.60), 25: 1.42 (1.90), and 36: 1.09 (1.20) for the PSM+ (*Supplementary file 1*)

and 9: 2.20 (2.13), 16: 2.88 (2.28), 25: 2.64 (2.57), and 36: 2.38 (2.22) for the GLAM+ (*Supplementary file 3*). Interestingly, we did not find any evidence for additive gaze biases in the IAM+ (with mean [s.d.] $\zeta$ estimates of 9: 0.23 [0.69], 16: 0.20 [0.73], 25: 0.32 [0.98], and 36: 0.31 [0.72]; *Supplementary file 2*), while multiplicative gaze biases were more pronounced (as indicated by mean [s.d.] $\gamma$ estimates of 9: 0.01 [0.05], 16: 0.005 [0.008], 25: 0.02 [0.05], and 36: 0.02 [0.09]; *Supplementary file 2*). However, the IAM+ generally also provided the worst fit to individuals' association of gaze and choice (*Figure 6H*).

We further studied the correlation between individual $\gamma$ and $\zeta$ estimates to better understand the association of multiplicative and additive gaze biases (*Figure 7*). In the PSM+ and GLAM+, $\gamma$ estimates generally decreased with increasing $\zeta$ estimates ($\beta$ = −0.06, 94% HDI = [−0.09, −0.03] per unit increase in $\zeta$ for the PSM+ (*Figure 7A*); $\beta$ = −0.08, 94% HDI = [−0.10, −0.05] per unit increase in $\zeta$ for the GLAM+ (*Figure 7C*); the mixed-effects regressions included a random slope and intercept for each set size), such that subjects with a stronger additive gaze bias (as indicated by larger $\zeta$ estimates) generally also exhibited stronger multiplicative gaze biases (as indicated by smaller $\gamma$ estimates). We did not find evidence for this association in the IAM+ ($\beta$ = 0.0003, 94% HDI = [−0.012, 0.014] per unit increase in $\zeta$; *Figure 7B*). Yet, the $\gamma$ and $\zeta$ estimates in the IAM+ generally also exhibited only little variability (see *Figure 5—figure supplement 1Q–R*).

## Discussion

The goal of this work was to identify the computational mechanisms underlying choice behaviour in MAFC, by comparing a set of decision models on choice, RT, and gaze data. In particular, we tested models of optimal and satisficing choice (*Reutskaja et al., 2011*; *Caplin et al., 2011*; *Fellows, 2006*; *Fellner et al., 2009*; *McCall, 1970*; *Payne, 1976*; *Schwartz et al., 2002*; *Stüttgen et al., 2012*) as well as relative (*Krajbich and Rangel, 2011*; *Thomas et al., 2019*) and independent evidence accumulation (*Smith and Vickers, 1988*). We further tested two variants of these models, with and without influences of gaze on subjective value. We found that subjects' behaviour qualitatively could not be explained by optimal choice or standard instantiations of satisficing. After incorporating active effects of gaze into a probabilistic version of satisficing, it explained the data well, slightly outperforming the evidence accumulation models in fitting choice and RT data. Yet, the relative accumulation model with active gaze influences provided by far the best fit to the observed association between gaze allocation and choice behaviour, which was not explicitly accounted for in the likelihood-based model comparison, thus demonstrating that gaze-driven relative evidence accumulation provides the most comprehensive account of behaviour in MAFC.

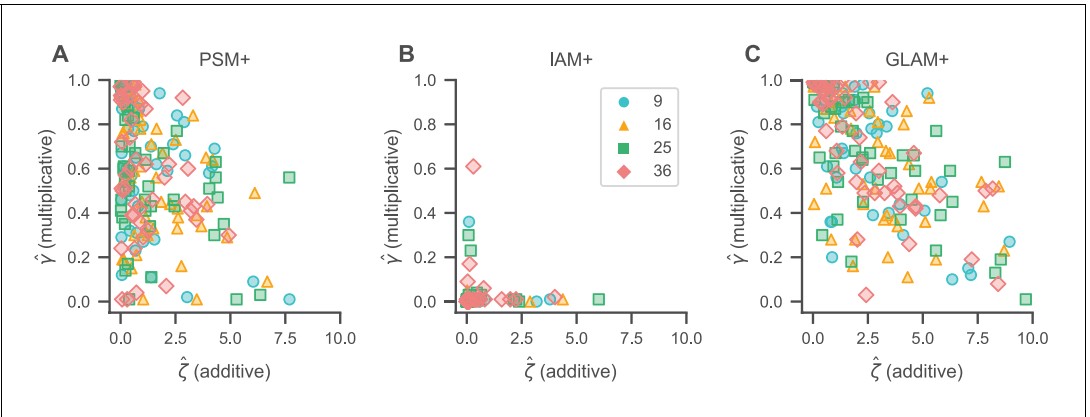

**Figure 7.** Gaze bias estimates. Association of individual $\gamma$ (multiplicative) and $\zeta$ (additive) gaze bias estimates for the PSM+ (**A**), IAM+ (**B**), and GLAM+ (**C**). $\gamma$ estimates generally decrease with increasing $\zeta$ estimates for the PSM+ (**A**) and GLAM+ (**C**), indicating that individuals with stronger additive gaze biases generally also exhibit stronger multiplicative gaze biases. We did not find any association of $\gamma$ and $\zeta$ estimates for the IAM+ (**B**). See the Gaze bias mechanism section for the corresponding statistical analyses. Colours and shapes represent different set sizes, while scatters indicate individual subjects.

One reason why the satisficing model might perform particularly well in capturing individuals' choices and RTs is that in our experiment there were a limited number of food items (80; see Materials and methods). Each item was repeated an average of 50 times per experiment. Thus, subjects could have learned to search for specific items. In practice, this strategy would only be useful in certain scenarios. At your local vending machine, you are almost guaranteed to encounter one of your favourite snacks; here satisficing would be useful. But at a foreign vending machine, or a new restaurant, the relative evidence accumulation framework might be more useful since you need to evaluate and compare the options. Future work is needed to investigate the performance of these models in novel and familiar choice environments.

These findings are also relevant to the discussion about the direction of causality between attention and choice. Several papers have argued that subjective value and/or the emerging choice affect gaze allocation, both in binary choice (*Cavanagh et al., 2014*; *Westbrook et al., 2020*) and in multi-alternative choice (*Krajbich and Rangel, 2011*; *Towal et al., 2013*; *Gluth et al., 2020*; *Callaway et al., 2020*). Other work has argued that gaze drives choice outcomes, using exogenous manipulations of attention (*Armel et al., 2008*; *Milosavljevic et al., 2012*; *Pärnamets et al., 2015*; *Tavares et al., 2017Gwinn et al., 2019*, c.f., *Newell and Le Pelley, 2018*; *Ghaffari and Fiedler, 2018*). Here, we find support for both directions of the association of gaze and choice. In contrast to the binary choice setting (*Krajbich et al., 2010*), we found that the probability that an item was looked at, as well as the duration of a gaze to this item, increased with the item's rating, and that this trend also increased over the course of a trial (*Figures 2* and *3*). Nevertheless, our data also indicate that gaze correlates with choice even after controlling for the ratings of the items (*Figure 4E,F*). This means that either gaze is driving choice or gaze is actively being allocated to items that are more appealing than normal, within a particular trial.

In a sense, the contrast between binary and multi-alternative choice is not surprising. When deciding between two alternatives, you are merely trying to compare one to the other. In that case, attending to either alternative is equally useful in reaching the correct decision. However, with many choice alternatives, it is in your best interest to quickly identify the best alternatives in the choice set and exclude all other alternatives from further consideration (e.g., *Hauser and Wernerfelt, 1990*; *Payne, 1976*; *Reutskaja et al., 2011*; *Roberts and Lattin, 1991*). Given this search and decision process, we might expect that subjects' choices are more driven by their gaze in the later stages of the decision, when they focus more on the highly rated items in the choice set, than in the earlier stages of the search, when gaze is driven by the items' positions and sizes. Indeed, we found that only the items' ratings predicted choice behaviour, not their positions or sizes.

Our modelling of the decision process included two types of gaze biases: an additive gaze bias, which increases the value of the currently looked at item by a constant, and a multiplicative gaze bias, which discounts the value of currently unattended items. As our experiment included only appetitive snack foods, both of these types of gaze biases have similar effects on the decision process, by increasing the subjective value of the currently looked-at alternative relative to the others. If, however, our experiment had included aversive snack foods, or the framing of the decision problem had been reversed (*Sepulveda et al., 2020*), we would have expected gaze to have the opposite effect. There is some evidence that in the former case, additive and multiplicative biases have opposite effects (*Westbrook et al., 2020*). More research is needed to better understand the interplay of these two types of gaze biases in choice situations that involve appetitive and aversive choice options.

Overall, our findings firmly reject a model of complete search and maximization in MAFC (*Caplin et al., 2011*; *Pieters and Warlop, 1999*; *Reutskaja et al., 2011*; *Simon, 1959*; *Stüttgen et al., 2012*): Subjects do not look at every item, and they do not always choose the best item they have seen. Our data also clearly reject the hard satisficing model: Subjects choose the last item they look at only half of the time. Additionally, we find that subjects' choices are strongly dependent on the actual time that they spend looking at each alternative and can therefore not be fully explained by simply accounting for the set of examined items. This stands in stark contrast to many models of consumer search and rational inattention (e.g., *Caplin et al., 2019*; *Masatlioglu et al., 2012*; *Matějka and McKay, 2015*; *Sims, 2003*), which ascribe a more passive role to visual attention, by viewing it as a filter that creates consideration sets (by attending only to a subset of the available alternatives) from which the decision maker then chooses. Our findings indicate that attention takes a much more active role in MAFC by guiding preference formation within

the consideration set, as has been observed with smaller choice sets (e.g., *Armel et al., 2008*; *Gluth et al., 2020*; *Krajbich et al., 2010*; *Krajbich and Rangel, 2011*; *Smith and Krajbich, 2019*; *Thomas et al., 2019*).

In conclusion, we find that models of gaze-weighted subjective value account for relations between eye-tracking data and choice that other passive-attention models of MAFC cannot. These findings provide new insight into the mechanisms underlying search and choice behaviour and demonstrate the importance of employing choice-process techniques and computational models for studying decision-making.

## Materials and methods

### Experimental design

Forty-nine healthy English speakers completed this experiment (17 females; 18–55 years, median: 23 years). All subjects were required to have normal or corrected-to-normal vision. Individuals wearing glasses or hard contact lenses were excluded from this study. Furthermore, individuals were only allowed to participate if they self-reportedly (1) fasted at least 4 hr prior to the experiment, (2) regularly ate the snack foods that were used in the experiment, (3) neither had any dietary restrictions nor (4) a history of eating disorders, and (5) did not diet within the 6 months prior to the experiment. The sample size for this experiment was determined based on related empirical research at the time of data collection (e.g., *Berkowitsch et al., 2014*; *Cavanagh et al., 2014*; *Krajbich et al., 2010*; *Krajbich and Rangel, 2011*; *Philiastides and Ratcliff, 2013*; *Reutskaja et al., 2011*; *Rodriguez et al., 2014*; *Towal et al., 2013*). Informed consent was obtained from all subjects in a manner approved by the Human Subjects Internal Review Board (IRB) of the California Institute of Technology (IRB protocol: 'Behavioural, eye-tracking, and psychological studies of simple decision-making'). Each subject completed the following tasks within a single session: First, they did some training with the choice task, followed by the choice task (*Figure 1*), a liking rating task, and the choice implementation.

In the choice task (*Figure 1*), subjects were instructed to choose the snack food item that they would like to eat most at the end of the experiment from sets of 9, 16, 25, or 36 alternatives. There was no time restriction on the choice phase and subjects indicated the time point of choice by pressing the space bar of a keyboard in front of them. After pressing the space bar, subjects had 3 s to indicate their choice with the mouse cursor (for an overview of the choice indication times, defined as the time difference between the space bar press and the click on an item image, see *Figure 1— figure supplement 3*). Subjects used the same hand to press the space bar and navigate the mouse cursor. If they did not choose in time, the choice screen disappeared and the trial was marked invalid and excluded from the analysis as well as the choice implementation. We further excluded trials from the analysis if subjects either chose an item that they did not look at before pressing the space-bar or if they clicked on the empty space between item images. The average number of trials dropped from the analysis was 4 (SE: 0.6) per subject and set size condition.

The initial training task had the exact same structure as the main choice task and differed only in the number of trials (five trials per set size condition) and the stimuli that were used (we used a distinct set of 36 snack food item images).

In the subsequent rating task, subjects indicated for each of the 80 snack foods, how much they would like to eat the item at the end of the experiment. Subjects entered their ratings on a 7-point rating scale, ranging from −3 (not at all) to 3 (very much), with 0 denoting indifference (for an overview of the liking rating distributions, see *Figure 1—figure supplements 1*, *2*).

After the rating task, subjects stayed for another 10 min and were asked to eat a single snack food item, which was selected randomly from one of their choices in the main choice task. In addition to one snack food item, subjects received a show-up fee of $10 and another $15 if they fully completed the experiment.

### Experimental stimuli

The choice sets of this experiment were composed of 9, 16, 25, or 36 randomly selected snack food images (random selection without replacement within a choice set). For each set size condition, these images were arranged in a square matrix shape, with the same number of images per row and

column (3, 4, 5, or 6). All images were displayed in the same size and resolution (205 × 133 px) and depicted a single snack food item centred in front of a consistent black background.

During the rating phase, single item images were presented one at a time and in their original resolution (576 × 432 px), again centred in front of a consistent black background (*Figure 1*). Overall, we used a set of 80 different snack food items for the choice task and a distinct set of 36 items for the training.

## Eye tracking

Monocular eye tracking data were collected with a remote EyeLink 1000 system (SR Research Ltd., Mississauga, Ontario, Canada), with a sampling frequency of 500 Hz. Before the start of each trial, subjects had to fixate a central fixation cross for at least 500 ms to ensure that they began each trial fixating on the same location (*Figure 1*).

Eye tracking measures were only collected during the choice task and always sampled from the subject's dominant eye (10 left-dominant subjects). Stimuli were presented on a 19-inch LCD display with a resolution of 1280 × 1024 px. Subjects had a viewing distance of about 50 cm to the eye tracker and 65 cm to the display. Several precautions were taken to ensure a stable and accurate eye tracking measurement throughout the experiment, as we presented up to 36 items on a single screen: (1) the eye tracker was calibrated with a 13-point calibration procedure of the EyeLink system, which also covers the screen corners, (2) four separate calibrations were run throughout the experiment: once before and after the training task and twice during the main choice task (after 75 and 150 trials), (3) subjects placed their head on a chin rest, while we recorded their eye movements.

Fixation data were extracted from the output files obtained by the EyeLink software package (SR Research Ltd., Mississauga, Ontario, Canada). We used these data to define whether the subject's gaze was either within a rectangular region of interest (ROI) surrounding an item (item gaze), somewhere else on the screen (non-item gaze), or whether the gaze was not recorded at all (missing gaze, e.g., eye blinks). All non-item and missing gazes occurring before the first and after the last gaze to an item in a trial were discarded from all gaze analyses. All missing data that occurred between gazes to the same item were changed to that item and thereby included in the analysis. A gaze pattern of 'item 1, missing data, item 1' would therefore be changed to 'item 1, item 1, item 1'. Non-item or missing gaze times that occurred between gazes to different items, however, were discarded from all gaze analyses.

## Item attributes

### Liking rating

An item's liking rating (or value) is defined by the rating that the subject assigned to this item in the liking rating task (*Figure 1*).

### Position

This metric described the position of an item in a choice set and was encoded by two integer numbers: one indicating the row in which the item was located and the other indicating the respective column. Row and column indices ranged between one and the square root of the set size (as choice sets had a square shape, with the same number of rows and columns; *Figure 1*). Indices increased from left to right and top to bottom. For instance, in a choice set with nine items, the column indices would be 1, 2, 3, increasing from left to right, while the row indices would also be 1, 2, 3, but increase from top to bottom. The item in the top left corner of a screen would therefore have a row and column index of 1, whereas the item in the top right corner would have a row index of 1 and a column index of 3.

### Size

This metric describes the size of an item depiction with respect to the size of its image. In order to compute this statistic, we made use of the fact that all item images had the exact same absolute size and resolution. First, we computed the fraction of the item image that was covered by the consistent black background. Subsequently, we subtracted this number from one to get a percentage estimate

of how much image space is covered by the snack food item. As all item images had the same size and resolution, these percentage estimates are comparable across images.

## Choice models

| | Passive gaze | Active gaze |
|---|---|---|
| *Probabilistic satisficing (PSM)* | **Stopping rule:** The probability $q(t)$ that individuals stop their search and make a choice increases with cumulative time $t$ and cached value $C(t)$ (scaled by $v$ and $\alpha$, respectively): $q(t) = v \times t + \alpha \times C(t)$ The cached value $C(t)$ represents the highest item value $l$, from the set of seen items $J$, that has been seen up to time $t$: $C(t) = max_{j \in J} \ l_j$ **Choice rule:** Once the search ends, individuals make a probabilistic choice (with scaling parameter $\tau$) ) over the set of seen items $J$ according to their liking values: $p_i = \frac{exp(\tau \times l_i)}{\sum_{j \in J} exp(\tau \times l_j)}$ | **Stopping rule:** The probability $q(t)$ that individuals stop their search and make a choice increases with cumulative time $t$ and cached value $C(t)$ (scaled by $v$ and $\alpha$, respectively): $q(t) = v \times t + \alpha \times C(t)$ Importantly, the cached value $C(t)$ is extended by the influence of gaze allocation: $C(t) = max_{j \in J} \ c_j(t)$ $c_i(t) = g_i(t) \ \times (l_i + \zeta) + (1 - g_i(t)) \ \times \gamma \times l_i$ $J$ represents the set of so far seen items, $g_i(t)$ indicates the fraction of cumulative time $t$ that an item $i$ has been looked at, while $\gamma$ and $\zeta$ determine the strength of the multiplicative and additive gaze bias, and $l_i$ indicates the item's value. **Choice rule:** Once the search ends, individuals make a probabilistic choice (with scaling parameter $\tau$) over the set of seen items $J$ according to the gaze-weighted values $c_i(t)$: $p_i(t) = \frac{exp(\tau \times c_i(t))}{\sum_{j \in J} exp(\tau \times c_j(t))}$ |
| *Independent evidence accumulation (IAM)* | The decision follows a stochastic evidence accumulation process, with one accumulator per item. Evidence accumulation for an item only starts once it is looked at in the trial. The drift rate $D_i$ of evidence accumulator $i$ is determined by the item's value $l_i$: $D_i = l_i$ | The decision follows a stochastic evidence accumulation process, with one accumulator per item. Evidence accumulation for an item only starts once it is looked at in the trial. The drift rate $D_i$ of evidence accumulator $i$ is defined as: $D_i = g_i \ \times (l_i + \zeta) + (1 - g_i) \times \ \gamma \times l_i$ $g_i$ represents the fraction of the remaining trial time that item $i$ has been looked at, after it is first seen in the trial, while $l_i$ indicates the item's value. $\gamma$ and $\zeta$ determine the strength of the multiplicative and additive gaze bias. |
| *Relative evidence accumulation (GLAM)* | The decision follows an evidence accumulation process, with one accumulator per item. We define the relative evidence as the difference between the item's value $l_i$ and the maximum value of all other seen items $J$. Importantly, evidence is only accumulated for an item if it is looked at in the trial. The drift rate $D_i$ of accumulator $i$ is defined as: $D_i = \sigma(l_i - max_{j \neq i} \ l_j)$ $\sigma(x) = \frac{1}{1 + exp(-\tau \times x)}$ $\tau$ indicates the scaling parameter of the logistic function $\sigma$. | A variant of the GLAM in which the absolute decision signal $A_i$ for item $i$ is extended by an additive gaze bias: $A_i = g_i \times \ (l_i + \zeta) + \ (1 - g_i) \times \gamma \times l_i$ $g_i$ represents the fraction of trial time that item $i$ has been looked at, while $l_i$ indicates its value. $\gamma$ and $\zeta$ determine the strength of the multiplicative and additive gaze bias. The drift rate $D_i$ of accumulator $i$ is defined as: $D_i = \sigma(A_i - max_{j \neq i} \ A_j)$ $\sigma(x) = \frac{1}{1 + exp(-\tau \times x)}$ $\tau$ represents the scaling parameter of the logistic function $\sigma$. |

## Probabilistic satisficing model

Our formulation of the PSM is based on a proposal by *Reutskaja et al., 2011* and consists of two distinct components: a probabilistic stopping rule, defining the probability $q(t)$ with which the search ends and a choice is made at each time point $t$ ($\Delta t = 1$ ms), and a probabilistic choice rule, defining a choice probability $\lambda_i$ for each item $i$ in the choice set. *Reutskaja et al., 2011* defined the stopping probability $q(t)$ as:

$$q(t) = min\{\alpha \times C(t) + v \times t, 1\} \text{ with } q(0) = 0, \ 0 < C(t), \ 0 \leq \{t, \alpha, v\} \qquad (1)$$

Importantly, $q(t)$ increases linearly with the cached item value $C(t)$ and trial time $t$. Note that we extend the original formulation of the model by *Reutskaja et al., 2011* upon an active influence of gaze on the decision process. Specifically, we defined the cached item value $C(t)$ as:

$$C(t) = max_{j \in J} \, c_j(t) \tag{2}$$

$$c_i(t) = g_i(t) \times (l_i + \zeta) + (1 - g_i(t)) \times \gamma \times l_i \tag{3}$$

Here, $J$ represents the set of items seen so far, while $g_i(t)$ indicates the fraction of elapsed trial time $t$ that item $i$ was looked at, and $\gamma$ $(0 \leq \gamma \leq 1)$ and $\zeta$ $(0 \leq \zeta \leq 10)$ implement the multiplicative and additive gaze bias effects. While an item $i$ is looked at, its value $l_i$ (as indicated by the item's liking rating) is increased by $\zeta$, whereas the value of all other items that are momentarily not looked at is discounted by $\gamma$. Note that we set $c_i(t) = 0$ for all items that were not yet looked at by time point $t$. To further ensure $0 < C(t)$, we re-scaled all liking ratings to a range from 1 to 7. The strength of the influence of $C(t)$ and $t$ on $q(t)$ is determined by the two positive linear weighting parameters $\alpha$ and $v$. Note that $q(t)$ is bounded to $0 \leq q(t) \leq 1$. To obtain the passive-gaze variant of the PSM, we set $\gamma = 1$ and $\zeta = 0$.

However, $q(t)$ does not account for the probability that the search has ended at any time point prior to $t$. In order to apply and fit the model to RT data, we need to compute the joint probability $f(t)$ that the search has not stopped prior to $t$ and the probability that the search ends at time point $t$. Therefore, we correct $q(t)$ for the probability $Q(t)$ that the search has not stopped at any time point prior to $t$:

$$f(t) = q(t) \times Q(t - 1) \tag{4}$$

$$Q(t) = \prod_{1}^{t} (1 - q(t)) \tag{5}$$

Once the search has ended, the model makes a probabilistic choice over the set of seen items $J$ following a softmax function of their cached values $c_i(t)$ (with scaling parameter $\tau$):

$$\sigma_i(t) = \frac{exp(\tau \times c_i(t))}{\sum_{j \in J} exp(\tau \times c_j(t))} \tag{6}$$

Lastly, by multiplying the stopping probability $f(t)$ by $\sigma_i(t)$, we obtain the probability $p_i(t)$ that item $i$ is chosen at time point $t$:

$$p_i(t) = f(t) \times \sigma_i(t) \tag{7}$$

## Independent evidence accumulation model

The IAM assumes that the choice follows an evidence accumulation process, in which evidence for an item is only accumulated once it was looked at in a trial and is then independent of all other items in a choice set (much like deciding whether the item satisfies a reservation value) (**Smith and Vickers, 1988**). A choice is determined by the first accumulator that reaches a common pre-defined decision boundary $b$, which we set to 1. Specifically, the evidence accumulation process is guided by a set of decision signals $D_i$ for each item $i$ that was looked at in the trial:

$$D_i = g_i \times (l_i + \zeta) + (1 - g_i) \times \gamma \times l_i \tag{8}$$

Here, $l_i$ indicates the item's value (as indicated by its liking rating), while $g_i$ indicates the fraction of the remaining trial time (after time point $t0_i$ at which item $i$ was first looked at in the trial) that the individual spent looking at item $i$. As in the PSM (**Equation 3**), $\gamma$ $(0 \leq \gamma \leq 1)$ and $\zeta$ $(0 \leq \zeta \leq 10)$ indicate the strength of the multiplicative and additive gaze bias. To obtain the passive-gaze variant of the IAM, we set $\gamma = 1$ and $\zeta = 0$.

At each time step $t$ (with $\Delta t = 1$ ms), the amount of accumulated evidence $E_i$ is determined by a velocity parameter $v$, the item's decision signal $D_i$, and zero-centred normally distributed noise with standard deviation $\sigma$:

$$E_i(t) = E_i(t - 1) + v \times D_i + N(0, \sigma^2) \text{ with } E_i(t < t0_i) = 0 \tag{9}$$

As for the PSM, we re-scaled all liking ratings to a range from 1 to 7 to ensure $0 < D_i$. Note that we set $E_i(t < t0_i) = 0$ for all items that were not yet looked at by time point t.

Lastly, the first passage time density $f_i(t)$ of a single linear stochastic accumulator $E_i(t)$ at time point t is given by the Inverse Gaussian Distribution (*Wald, 2004*):

$$f_i(t) = \left[\frac{\lambda}{2\pi t^3}\right]^{1/2} \times exp\left\{\frac{-\lambda \times (t-\mu)^2}{2\mu^2 \times t}\right\} \text{ with } \mu = \frac{b}{v \times D_i} \text{ and } \lambda = \frac{b^2}{\sigma^2}. \tag{10}$$

With cumulative distribution function $F_i(t)$:

$$F_i(t) = \Phi\left(\sqrt{\frac{\lambda}{t}} \times \left(\frac{t}{\mu} - 1\right)\right) + exp\left(\frac{2\lambda}{\mu}\right) \times \Phi\left(-\sqrt{\frac{\lambda}{t}} \times \left(\frac{t}{\mu} + 1\right)\right), \tag{11}$$

where $\Phi$ is the standard normal cumulative distribution function.

However, $f_i(t)$ does not take into account that there are multiple accumulators in each trial racing towards the same decision boundary. A choice is made as soon as any of these accumulators reaches the boundary. We thus correct $f_i(t)$ for the probability that any other accumulator $j$ crosses the boundary first, thereby obtaining the joint probability $p_i(t)$ of an accumulator reaching the boundary at the empirically observed RT, and no other accumulator $j$ having reached it prior to RT:

$$p_i(RT) = f_i(RT - t0_i) \times \prod_{j \neq i}\left(1 - F_j\left(RT - t0_j\right)\right) \tag{12}$$

## Gaze-weighted linear accumulator model

The GLAM (*Thomas et al., 2019*; *Molter et al., 2019*) assumes that choices are driven by the accumulation of noisy evidence in favour of each available choice alternative i. As for the IAM (*Equations 8–12*), a choice is determined by the first accumulator that reaches a common pre-defined decision boundary b ($b = 1$). Particularly, the accumulated evidence $E_i$ in favour of alternative i is defined as a stochastic process that changes at each point in time $t$ ($\Delta t = 1ms$) according to:

$$E_i(t) = E_i(t-1) + v \times D_i + N\left(0, \sigma^2\right) \text{ with } E_i(0) = 0 \tag{13}$$

$E_i(t)$ consists of a drift term $D_i$ and zero-centred normally distributed noise with standard deviation $\sigma$. Note that we only included choice alternatives in the decision process that were also looked at in a trial (by setting $E_i(t) = 0$ for all other alternatives). The overall speed of the accumulation process is determined by the velocity parameter $v$. The drift term $D_i$ of accumulator i is defined by a set of decision signals: an absolute and a relative decision signal. The absolute decision signal implements the model's gaze bias mechanism. Importantly, the variant of the GLAM used here extends the gaze bias mechanism of the original GLAM to include an additive influence of gaze on the decision process (in line with recent empirical findings; *Cavanagh et al., 2014*; *Westbrook et al., 2020*). The absolute decision signal can thereby be in two states: An additive state, in which the item's value $l_i$ (as indicated by the item's liking rating), is amplified by a positive constant $\zeta$ ($0 \leq \zeta \leq 10$), while the item is looked at, and a multiplicative state, while any other item is looked at, where the item value $l_i$ is discounted by $\gamma$ ($0 \leq \gamma \leq 1$). The average absolute decision signal $A_i$ is then given by

$$A_i = g_i \times (l_i + \zeta) + (1 - g_i) \times \gamma \times l_i \tag{14}$$

Here, $g_i$ describes the fraction of total trial time that the decision maker spends looking at item i. To obtain the passive-gaze variant of the GLAM, we set $\gamma = 1$ and $\zeta = 0$.

We define the relative decision signal $R_i$ of item i as the difference in the average absolute decision signal $A_i$ and the maximum of all other absolute decision signals J:

$$R_i = A_i - max_{j \neq i} A_j \tag{15}$$

The GLAM further assumes that the decision process is particularly sensitive to differences in the relative decision signals $R_i$ which are close to 0 (where the average absolute decision signal $A_i$ for an item i is close to the maximum of all other items J). To account for this, the GLAM scales the relative decision signals $R_i$ by the use of a logistic transform $\sigma$ with scaling parameter $\tau$:

$$D_i = \sigma(R_i) \tag{16}$$

$$\sigma(x) = \frac{1}{1 + exp(-\tau \times x)} \tag{17}$$

This transform also ensures that the drift terms $D_i$ of the stochastic race are positive, whereas the relative decision signals $R_i$ can be positive and negative (*Equation 15*).

Similar to the IAM (*Equations 10–12*), we can obtain the joint probability $p_i(t)$ of an accumulator reaching the boundary at time *t*, and no other accumulator *j* having reached it prior to *t*, as follows:

$$p_i(t) = f_i(t) \times \prod_{j \neq i} \left(1 - F_j(t)\right) \tag{18}$$

Note that *f(t)* and *F(t)* follow *Equations 10 and 11*.

## Parameter estimation

All model parameters were estimated separately for each individual in each set size condition. The individual models were implemented in the Python library PyMC3.9.1 (*Salvatier et al., 2016*) and fitted using Markov chain Monte Carlo Metropolis sampling. For each model, we first sampled 5000 tuning samples that were then discarded (burn-in), before drawing another 5000 additional posterior samples that we used to estimate the model parameters. Each parameter trace was checked for convergence by means of the Gelman–Rubin statistic ($|R - 1| < 0.05$) as well as the mean number of effective samples (>100). If a trace did not converge, we re-sampled the model and increased the number of burn-in samples by 5000 until convergence was achieved. Note that the IAM+ did not converge for three, one, and one subjects in the set sizes with 9, 16, and 25 items, respectively, after 50 re-sampling attempts. For these subjects, we continued all analyses with the model that was sampled last. We defined all model parameter estimates as maximum a posteriori estimates (MAP) of the resulting posterior traces (for an overview, see *Figure 5—figure supplement 1* and *Supplementary files 1–3*).

### Probabilistic satisficing model

The PSM has five parameters, which determine the additive ($\zeta$) and multiplicative ($\gamma$) gaze bias, the influence of cached value ($\alpha$) and time ($v$) on its stopping probability, and the sensitivity of its softmax choice rule ($\tau$). We placed uninformative, uniform priors on all model parameters:

- $\zeta$ ~ Uniform(0, 10)
- $\gamma$ ~ Uniform(0, 1)
- $v$ ~ Uniform(0, 0.001)
- $\alpha$ ~ Uniform(0, 0.001)
- $\tau$ ~ Uniform(0, 10)

### Independent evidence accumulation model

The iIAM has four parameters, which determine its general accumulation speed ($v$) and noise ($\sigma$) and its additive ($\zeta$) and multiplicative ($\gamma$) gaze bias. We placed uninformative, uniform priors on all model parameters:

- $v$ ~ Uniform(1e-7, 0.005)
- $\sigma$ ~ Uniform(1e-7, 0.05)
- $\zeta$ ~ Uniform(0, 10)
- $\gamma$ ~ Uniform(0, 1)

### Gaze-weighted linear accumulator model

The GLAM variant used here has five parameters, which determine its general accumulation speed ($v$) and noise ($\sigma$), its additive ($\zeta$) and multiplicative ($\gamma$) gaze bias, and the sensitivity of the scaling of the relative decision signals ($\tau$). We placed uninformative, uniform priors between on all model parameters:

- $v \sim$ Uniform(1e-7, 0.005)
- $\sigma \sim$ Uniform(1e-7, 0.05)
- $\zeta \sim$ Uniform(0, 10)
- $\gamma \sim$ Uniform(0, 1)
- $\tau \sim$ Uniform(0, 10)

## Error likelihood model

In line with existing DDM toolboxes (e.g., *Wiecki et al., 2013*), we include spurious trials at a fixed rate of 5% in all model estimation procedures (*Equation 20*). We model these spurious trials with a subject-specific uniform likelihood distribution $u_s$. This likelihood describes the probability of a random choice for any of the $N$ available items at a random time point in the range of a subject's empirically observed response times $RT_s$ (*Ratcliff and Tuerlinckx, 2002*):

$$u_s(t) = \frac{1}{N \times (max\,RT_s - min\,RT_s)} \tag{19}$$

The resulting likelihood $l_{s,i}(t)$ of subject $s$ choosing item $i$ at time $t$ for all estimated models was thereby given by:

$$l_{s,i}(t) = 0.95 \times p_i(t) + 0.05 \times u_s(t) \tag{20}$$

## Model simulations

We repeated each trial 50 times during the simulation and simulated a choice and RT for each trial with each model at a rate of 95%, while we simulated random choices and RTs according to *Equation 19* at a rate of 5%. We used the MAP of the posterior traces of the individual subject models as parameter estimates for the simulation (for an overview, see *Figure 5—figure supplement 1* and *Supplementary files 1–3*).

### Probabilistic satisficing model

For each trial repetition, we simulated a choice and response time according to *Equations 1 and 6*.

### Independent evidence accumulation model

For each trial repetition, we first drew a first passage time ($FPT_i$) for each item $i$ in a choice set according to *Equation 10*. To also account for the gaze-dependent onsets of evidence accumulation, we then added the empirically observed time at which the item was first looked at in the trial ($t0_i$, see *Equations 9 and 12*) to the drawn $FPT_i$ of each item. The item with the shortest $FPT_i + t0_i$ then determined the simulated trial choice and RT.

### Gaze-weighted linear accumulator model

For each trial repetition, we simulated a choice and response time by drawing a first passage time ($FPT_i$) for each item in a choice set according to *Equation 12*. The item with the shortest $FPT_i$ determined the RT and choice.

## Mixed-effects modelling

All mixed-effects models were fitted in a Bayesian hierarchical framework by the use of the Bayesian Model-Building Interface (bambi 0.2.0; *Capretto et al., 2021*). Bambi automatically generates weakly informative priors for all model variables. We fitted all models using the Markov chain Monte Carlo No-U-Turn-Sampler (*Hoffman and Gelman, 2014*), by drawing 2000 samples from the posterior, after a minimum of 500 burn-in samples. In addition to the reported fixed-effect estimates, all models included random intercepts for each subject, as well as random subject-slopes for each model coefficient. The posterior traces of all reported fixed-effects estimates were checked for convergence by means of the Gelman–Rubin statistic ($|R - 1| < 0.05$). If a fixed-effect posterior trace did not converge, the model was re-sampled and the number of burn-in samples increased by 2000 until convergence was achieved.

## Software

All data analyses were performed in Python 3.6.8 (Python Software Foundation), by the use of the SciPy 1.3.1, (*Virtanen et al., 2019*), NumPy 1.17.3 (*Oliphant, 2006*), Matplotlib 3.1.1 (*Hunter, 2007*), Pandas 0.25.2 (*McKinney, 2010*), Theano 1.0.4 (*The Theano Development Team, 2016*), bambi 0.2.0 (Yarkoni & Westfall, 2016), ArviZ 0.9.0 (*Kumar et al., 2019*), and PyMC3.9.1 (*Salvatier et al., 2016*) packages. For the computation of stimulus metrics, we further utilized the Pillow 5.0 (http://pillow.readthedocs.io) Python package. The experiment was written in MATLAB (The MathWorks, Inc, Natick, MA), using the Psychophysics Toolbox extensions (*Brainard, 1997*).

## Availability of data, model, and analysis code

All experiment stimuli, data, and analysis scripts are available at: https://github.com/athms/many-item-choice (*Thomas, 2021*; copy archived at swh:1:rev: 7b8d6d852f89ad0e59ace94614acc6b683d914e0).

# Acknowledgements

Armin W Thomas is supported by the Max Planck School of Cognition and Stanford Data Science. Felix Molter is supported by the International Max Planck Research School on the Life Course (LIFE). Ian Krajbich is funded by National Science Foundation Career Award 1554837 and the Cattell Sabbatical Fund. All data were collected in the Rangel Neuroeconomics Laboratory. Antonio Rangel was also involved in conceiving the experiment. Thanks to Wenjia Zhao for comments on the manuscript.

# Additional information

## Funding

| Funder | Grant reference number | Author |
| --- | --- | --- |
| National Science Foundation | 1554837 | Ian Krajbich |
| Cattell Sabbatical Fund | | Ian Krajbich |
| Max Planck School of Cognition | | Armin W Thomas |
| WZB Berlin Social Science Center | | Felix Molter |

The funders had no role in study design, data collection and interpretation, or the decision to submit the work for publication.

## Author contributions

Armin W Thomas, Conceptualization, Data curation, Software, Formal analysis, Investigation, Visualization, Methodology, Writing - original draft, Writing - review and editing; Felix Molter, Software, Methodology, Writing - review and editing; Ian Krajbich, Supervision, Investigation, Methodology, Writing - original draft, Project administration, Writing - review and editing

## Author ORCIDs

Armin W Thomas 🔹 https://orcid.org/0000-0002-9947-5705
Felix Molter 🔹 https://orcid.org/0000-0002-3283-1090
Ian Krajbich 🔹 https://orcid.org/0000-0001-6618-5675

## Ethics

Human subjects: Informed consent was obtained from all subjects in a manner approved by the Human Subjects Internal Review Board (IRB) of the California Institute of Technology (IRB protocol title: "Behavioural, eye-tracking, and psychological studies of simple decision-making", Protocol # 14-021).

**Decision letter and Author response**
Decision letter https://doi.org/10.7554/eLife.57012.sa1
Author response https://doi.org/10.7554/eLife.57012.sa2

## Additional files

### Supplementary files

• Supplementary file 1. Mean parameter estimates of the probabilistic satisficing model with active (PSM+) and passive (PSM) account of gaze in the decision process for each set size. The probabilistic satisficing model has five parameters, determining the additive ($\zeta$) and multiplicative ($\gamma$) gaze bias effects on its cached value, the influence of cached value ($\alpha$) and time ($v$) on its stopping probability, and the sensitivity of its softmax choice rule ($\tau$). Note that the high mean value of $\alpha$ for the active-gaze variant in the set size with 16 items is driven by one outlier (*Figure 5—figure supplement 1D*).

• Supplementary file 2. Mean parameter estimates of the independent evidence accumulation model with active (IAM+) and passive (IAM) account of gaze in the decision process for each set size. The independent evidence accumulation model has four parameters, determining its additive ($\zeta$) and multiplicative ($\gamma$) gaze bias effects and its general accumulation speed ($v$) and noise ($\sigma$).

• Supplementary file 3. Mean parameter estimates for the gaze-weighted linear accumulator model with active (GLAM+) and passive (GLAM) account of gaze in the decision process for each set size. The GLAM variant used in this work has five parameters, determining its additive ($\zeta$) and multiplicative ($\gamma$) gaze bias, its general accumulation speed ($v$) and noise ($\sigma$) as well as the sensitivity of the scaling of the relative decision signals ($\tau$).

• Transparent reporting form

### Data availability

All experiment stimuli, data, and analysis scripts are available at: https://github.com/athms/many-item-choice (copy archived at https://archive.softwareheritage.org/swh:1:rev:7b8d6d852f89ad0e59a-ce94614acc6b683d914e0/).

The following dataset was generated:

| Author(s) | Year | Dataset title | Dataset URL | Database and Identifier |
|---|---|---|---|---|
| Thomas AW, Molter F, Krajbich I | 2020 | Uncovering the computational mechanisms underlying many-alternative choice | https://github.com/athms/many-item-choice | Github, many-item-choice |

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
