## [Decision Letter]

**Acceptance summary:**

All the reviewers and the Reviewing Editor found this to be very powerful work that extends our knowledge of decision dynamics in two very important directions. First, this work provided a new and interesting understanding on the influence of gaze in decision making (a factor often ignored in previous work). Second, the evaluation of this gaze-dependent phenomenon in the context of high choice option conditions extends our work beyond the more standard 2 choice framework dominated by accumulation-to-bound models of decision making.

**Decision letter after peer review:**

Thank you for sending your article entitled "Uncovering the Computational Mechanisms Underlying Many-Alternative Choice" for peer review at *eLife*. Your article is being evaluated by three peer reviewers, and the evaluation is being overseen by Timothy Verstynen as the Reviewing Editor and Michael Frank as the Senior Editor.

All reviewers, and the Review Editor, were very enthusiastic about this study, but felt that it fell short of achieving its full potential. As you can see in the reviews below, all reviewers wanted greater clarity on the specificity of the mechanism of how gaze (or attention) influences multi-option decisions. However, there are multiple ways that this could be done: more exhaustive modeling or new experimental data or both.

Reviewer #1:

How do people choose their preferred item from a large set? Thomas et al. compare different models to explain choice behavior in such a task, ranging from extensions of classical 2AFC decision models that include a gaze-dependency (GLAM), to optimal and heuristic (satisficing) models originating in research on consumer choice. They find that choice behavior can be reasonably well explained by a form of probabilistic satisficing (termed the “hybrid” model). However, this model does not account for the finding that the longer people look at an item, the more likely they are to choose it. Thomas et al. thus propose that gaze-dependent accumulation is used to choose among many items.

The main question of this research is an important one: models of 2AFC are all too often tacitly assumed to capture decision-making behavior in more complex situations. The work is technically sound, well written, and clearly presented.

However, I find the authors' conclusions a bit vague, mostly due to the models they choose to compare: their “hybrid” and GLAM models differ in quite a few ways, and no attempt is made to bridge the space between them (for example, by allowing the “liking” value in the “hybrid” model to depend on gaze duration). While they seem to conclude that gaze-dependent accumulation beats a “hybrid” satisficing model, they only achieve this by adding in a term for elite items that follow a satisficing rule.

Overall, while I enjoyed reading this paper, I find the conclusions somewhat disappointing. A full understanding of choice behavior between many items warrants searching over a larger space of models that bridge the space between gaze-dependent accumulation and probabilistic “satisficing” stopping rules. At the very least, the authors should justify why they believe the four models they test are sufficient to span the space of possibilities, and why they do not explore hybrid models between the two winning ones.

Reviewer #2:

The authors compare two models of decision-making (hybrid optimal choice and satisficing) and a gaze-driven evidence accumulation model (GLAM) in many-alternative value-based choice with varying set sizes. They find that GLAM outperforms the Hybrid model, but decreasingly so as set size increases, which they connect to the probability of "elite" options, that will be chosen as soon as encountered and terminate the choice process. They argue that the superiority of the GLAM is further evidenced by its unique ability to capture the relationship between gaze and choice. The work addresses an interesting question and shows rigor in the execution. It demonstrates that the relevance of gaze/attention in choice extends to larger set sizes. I have one major concern related to the interpretation of the relationship between gaze and choice as unidirectional.

1) The authors suggest both in their writing as well as in their model specification (e.g. aDDM simulations) that gaze drives choice, but not vice versa. Recent evidence, as well as patterns in the authors' data (e.g. Figure 3C) suggest that gaze is not in fact independent of the choice process (cf. Gluth et al., 2020, Westbrook et al., 2020, Callaway and Griffith, 2019). To the least, this should be explicitly acknowledged and the wording adjusted accordingly, but given their emphasis on GLAM's superiority based on that very relationship, and the title's broad claim to elucidate the computational mechanisms underlying many-alternative choice, it seems even more appropriate to compare their model to one that takes choice-dynamic effects on gaze into account (e.g. Gluth et al., 2020). This is specifically important given that the hybrid model is not designed to account for gaze effects and therefore using this specific criterion for evaluation seems somewhat misplaced, particularly given the mixed results due to elite items that resulted in the new hybrid.

Reviewer #3:

Studying decisions without a deadline, and with many options, here 9, 16, 25 or 36, is needed for decision making as a field to move on from 2AFC. As is, in my opinion, considering multiple models, as hybrids. But the authors' focus on making some point about the benefits of their model, GLAM, means they miss their opportunity.

This work sets the stage for a great next paper. It offers a glimpse of new theoretical accounts, but I think this glimpse is for the specialist.

A note to the authors: Thank you for doing this work. This review was hard for me to write, though I am sure it is harder to read. I think the experiment is interesting, and useful, and I am happy you did it. Please don't be too discouraged.

– Is it gaze that matters, so gaze terms could be added to range of existing accounts, or your particular account of gaze, in GLAM?

- If you want to prove a point about GLAM then do model comparisons, as you are, but leave elite-GLAM out of it.

– Adding elite-GLAM, ad hoc, at the end, made everything confusing for me. I am not sure by the end what you are going for here.

– If you want to explore the broader theoretical space, that elite-GLAM is small part of, why focus on GLAM per se, and not on more on fuller range of hybrid and theoretical possibilities that are open to you, given we have this huge decision making literature.

– The behavioral analysis is interesting, and to me under-played. Looking into individual subjects differences, and spending the time to highlight consistencies with the averages would be worthwhile. This is new task and I don't leave the paper understanding its basic results as well I would like.

[Editors' note: further revisions were suggested prior to acceptance, as described below.]

Thank you for submitting your article "Uncovering the Computational Mechanisms Underlying Many-Alternative Choice" for consideration by *eLife*. Your article has been reviewed by two peer reviewers, and the evaluation has been overseen by a Reviewing Editor and Michael Frank as the Senior Editor. The reviewers have opted to remain anonymous.

The reviewers have discussed the reviews with one another and the Reviewing Editor has drafted this decision to help you prepare a revised submission.

Summary:

The authors provide a comprehensive characterization of decision-making in many alternative choice. They bridge research into decision-making with small choice sets with research into many alternative forced choice. These two literatures were previously surprisingly disparate in their perspectives, as well as their models of the choice processes. The authors show that combining insights from both literatures helps yield a more comprehensive understanding of how decisions are made. They also show that looking beyond the typical data helps resolve points of contention within lines of work (e.g. the relationship between gaze and choice).

The reviewers felt that the revised manuscript goes a long way to addressing the key points raised in the previous review. There is only one set of requested revisions that stem from the new model fits. These should only require a few small conceptual changes for clarity and linking to prior work.

Revisions:

1) New model fits

Reviewer #1 has one final concern with the manuscript, resulting from the new model fits; it does not become entirely clear how the reader should think about gaze-dependent choice behavior in this task. The PSM+ seems to be the best model overall, but cannot account for the specific relationship between choice and gaze; and the relative and independent evidence accumulation models each have their advantages depending on the set size. After reading, this leaves the reader wondering how tey should best think about the mechanisms of choice behavior in this task.

Now, the data are of course the data. We do not expect the authors to make overblown claims about one winning model, if this is not supported by the data. But a bit more interpretation and context throughout, as well as in the discussion, would help guide the reader; how do the authors think choice behavior most likely comes about? If they can't distinguish between some models (depending on the metric used), why does this conflict arise? Are there future modelling efforts or experiments that you can propose that would clarify this issue?

---

## [Author Response]

Reviewer #1:How do people choose their preferred item from a large set? Thomas et al. compare different models to explain choice behavior in such a task, ranging from extensions of classical 2AFC decision models that include a gaze-dependency (GLAM), to optimal and heuristic (satisficing) models originating in research on consumer choice. They find that choice behavior can be reasonably well explained by a form of probabilistic satisficing (termed the “hybrid” model). However, this model does not account for the finding that the longer people look at an item, the more likely they are to choose it. Thomas et al. thus propose that gaze-dependent accumulation is used to choose among many items.The main question of this research is an important one: models of 2AFC are all too often tacitly assumed to capture decision-making behavior in more complex situations. The work is technically sound, well written, and clearly presented.However, I find the authors' conclusions a bit vague, mostly due to the models they choose to compare: their “hybrid” and GLAM models differ in quite a few ways, and no attempt is made to bridge the space between them (for example, by allowing the “liking” value in the “hybrid” model to depend on gaze duration). While they seem to conclude that gaze-dependent accumulation beats a “hybrid” satisficing model, they only achieve this by adding in a term for elite items that follow a satisficing rule.Overall, while I enjoyed reading this paper, I find the conclusions somewhat disappointing. A full understanding of choice behavior between many items warrants searching over a larger space of models that bridge the space between gaze-dependent accumulation and probabilistic “satisficing” stopping rules. At the very least, the authors should justify why they believe the four models they test are sufficient to span the space of possibilities, and why they do not explore hybrid models between the two winning ones.

We thank the reviewer for this valuable remark. In line with this remark, we have identified a set of six models, which we compare in our revised manuscript. These models combine an either passive or active account of gaze in the decision process with three distinct accounts of the decision mechanism:

– The passive account of gaze assumes that gaze allocation solely determines the set of items that are considered in the decision process, without having any further influence on the probability that any of the looked-at items is chosen.

– The active account of gaze, on the other hand, assumes that gaze is driving the decision process with generally higher choice probabilities for items that were looked at longer. Importantly, two distinct hypotheses for the influence of gaze on the decision process have been reported in the literature: multiplicative effects of gaze on the decision process (e.g., Krajbich et al., 2010, 2011; Lopez-Persem, Domenech and Pessiglione, 2017; Tavares, Perona, Rangel, 2017; Smith and Krajbich, 2019; Thomas et al., 2019) as well as additive effects of gaze (cf., Cavanagh, Wiecki, Kochar and Frank, 2014; Westbrook et al., 2020). We specifically account for both of these factors in the modeling of the effect of gaze on the decision process.

In addition to these two accounts of gaze, we consider the following three accounts of the decision mechanism:

– Probabilistic satisficing (renamed from “hybrid model”, in response to a request by reviewer #1): The time point at which the individual stops their search and makes a choice follows a probabilistic stopping rule. Once the search process ends, the individual makes a probabilistic choice over the set of seen items.

– Independent evidence accumulation: The decision follows an evidence accumulation process, in which evidence for an item is accumulated independently from the other items in a choice set. A choice is made as soon as the accumulated evidence for an item reaches a predefined decision threshold.

– Relative evidence accumulation (as captured by the gaze-weighted linear accumulator model (GLAM)): The decision follows an evidence accumulation process, in which evidence for an item is accumulated relative to the other items in a choice set. A choice is made as soon as the accumulated evidence for an item reaches a predefined decision threshold.

Overall, this results in the three-by-two grid of models, see subsection “Choice models”

In a likelihood-based model comparison (see Figure 5 of the revised manuscript), we find that the models with an active account of gaze consistently outperform the model variants with a passive account of gaze. This indicates that gaze does more than bring an alternative into the consideration set, it actively increases the subjective value of the attended alternatives. Overall, the probabilistic satisficing model performs best at capturing individuals’ choices and response times, with the relative evidence accumulation model performing comparably well with 9 alternatives, but then falling behind for larger sets, and steadily losing ground to the independent evidence accumulation model as the set sizes increase.

We further tested the ability of the active-gaze model variants to capture choice behaviour on an absolute level, by first simulating choice and response time (RT) data for each subject with each model and then comparing the simulated and empirically observed data (see Figure 6 of the revised manuscript). We find that the probabilistic satisficing and relative evidence accumulation models accurately recover individuals’ mean RT, which the independent evidence accumulation model accurately, but imprecisely recovers. We further find that all three models underestimate high probabilities of choosing the highestrated item from a choice set, while the probabilistic satisficing model provides the overall best account of this metric, followed by the relative and independent evidence accumulation models. Turning to gaze data, we find that the relative evidence accumulation model best captures individuals’ association of gaze allocation and choice behaviour, while the probabilistic satisficing and independent evidence accumulation models generally struggle to recover strong associations of gaze and choice.

Lastly, we have entirely removed the elite-GLAM hypothesis from our revised manuscript. The purpose of the elite-GLAM hypothesis was to give the reader an intuition for how a satisficing and gaze-driven evidence accumulation process could interact in many-item choice situations. This is not necessary anymore in our revised manuscript, as we specifically design and compare a set of models spanning the space between gaze-driven evidence accumulation and satisficing.

Resulting main changes to the manuscript (excluding all model-related changes to the Materials and methods section):

“Here, we sought to study the mechanisms underlying MAFC, by developing and comparing different decision models on choice, response-time (RT), liking rating, and gaze data from a choice task with sets of 9, 16, 25, and 36 snack foods. […] Nevertheless, relative accumulation provides the overall best account of the empirically observed positive relation of gaze allocation and choice behaviour.”

“Based on the findings by Reutskaja and colleagues (2011), we also considered a probabilistic version of satisficing, which combines elements from the optimal choice and hard satisficing models. […] We allow for both of these mechanisms in the modeling of the active influence of gaze on the decision process (see Materials and methods).”

“Taken together, our findings have shown that subjects’ choice behaviour in MAFC does not match the assumptions of optimal choice or hard satisficing, while it qualitatively matches the assumptions of probabilistic satisficing and gaze-driven evidence accumulation. […] The GLAM+, in contrast, only slightly underestimated the association between gaze and choice (Figure 6 I; intercept = 3.01%, 94% HDI = [-5.79, -0.22], β = 0.86%, 94% HDI = [0.76, 0.96] per percentage increase in observed gaze influence).”

“The goal of this work was to identify the computational mechanisms underlying choice behaviour in MAFC, by comparing a set of decision models on choice, RT, and gaze data. […] Meanwhile, the number of decision processes in the independent accumulator model only grows linearly with the number of seen alternatives.”

Reviewer #2:The authors compare two models of decision-making (hybrid optimal choice and satisficing) and a gaze-driven evidence accumulation model (GLAM) in many-alternative value-based choice with varying set sizes. They find that GLAM outperforms the Hybrid model, but decreasingly so as set size increases, which they connect to the probability of "elite" options, that will be chosen as soon as encountered and terminate the choice process. They argue that the superiority of the GLAM is further evidenced by its unique ability to capture the relationship between gaze and choice. The work addresses an interesting question and shows rigor in the execution. It demonstrates that the relevance of gaze/attention in choice extends to larger set sizes. I have one major concern related to the interpretation of the relationship between gaze and choice as unidirectional.1) The authors suggest both in their writing as well as in their model specification (e.g. aDDM simulations) that gaze drives choice, but not vice versa. Recent evidence, as well as patterns in the authors' data (e.g. Figure 3C) suggest that gaze is not in fact independent of the choice process (cf. Gluth et al., 2020, Westbrook et al., 2020, Callaway and Griffith, 2019). To the least, this should be explicitly acknowledged and the wording adjusted accordingly, but given their emphasis on GLAM's superiority based on that very relationship, and the title's broad claim to elucidate the computational mechanisms underlying many-alternative choice, it seems even more appropriate to compare their model to one that takes choice-dynamic effects on gaze into account (e.g. Gluth et al., 2020). This is specifically important given that the hybrid model is not designed to account for gaze effects and therefore using this specific criterion for evaluation seems somewhat misplaced, particularly given the mixed results due to elite items that resulted in the new hybrid.

We thank the reviewer for this important remark, which addresses the causality of the association between gaze and choice.

We agree with the reviewer that this is a theoretically very important question, as it addresses the causality between the association of gaze and choice. Yet, we do not believe that formally modeling this phenomenon is within the scope of our manuscript. We do however document evidence for this bidirectional account.

Clearly, our data indicates that gaze allocation is being influenced by the subjective value of the items in a choice set (e.g., Figure 2A-H of our manuscript), as both the probability that individuals look at an item, as well as the duration of a gaze to the item, are strongly influenced by the item’s value. Previous research has already indicated this to some extent in three- and four-item choices (e.g., Krajbich and Rangel, 2011; Towal, Mormann and Koch, 2013; Gluth, Kern, Kortmann and Vitali, 2020). Nevertheless, our data also indicates that gaze is affecting choice above and beyond the values of the items in many-item choice (with generally increasing choice probabilities for items that were looked at longer; see for example, Figure 4E of our revised manuscript).

There are some ongoing efforts to incorporate a bi-directional association between gaze and choice into a single model of the decision process (Callaway, Rangel, Griffiths, *working paper;* Jang, Sharma, Drugowitsch, *working paper*), but this work is unpublished and highly complex (even for 3AFC). We feel it is premature to try to extend that work to our setting. However, we have now included an additive gaze effect to all the active-gaze models, in line with Westbrook et al., 2020.

We further want to highlight that we have deliberately avoided modeling any bi-directional accounts of gaze in our manuscript by utilizing the GLAM. The GLAM circumvents the complex problem of modeling individual gaze patterns, as it aggregates over the specific gaze trajectory and solely requires the empirically observed distribution of gaze times. Modeling individual gaze patterns in MAFC is highly complex and computationally intensive. This is especially true for our experiment, in which individuals can take as long as they want to look at the available choice set before making a decision.

To address this in our revised manuscript, we have extended our Discussion by adding a section that highlights the bi-directional relationship between gaze and choice. We hope that this will give the reader a deeper insight into this aspect of the data.

Reviewer #3:[…]- If you want to prove a point about GLAM then do model comparisons, as you are, but leave elite-GLAM out of it.– Adding elite-GLAM, ad hoc, at the end, made everything confusing for me. I am not sure by the end what you are going for here.

We thank the reviewer for this valuable remark. We entirely agree and have removed the elite-GLAM hypothesis from our revised manuscript. The purpose of the elite-GLAM hypothesis was to give the reader an intuition for how a satisficing and gaze-driven evidence accumulation process could interact in many-item choice situations. This is not necessary anymore in our revised manuscript, as we specifically design and compare a set of models spanning the space between gaze-driven evidence accumulation and satisficing.

– If you want to explore the broader theoretical space, that elite-GLAM is small part of, why focus on GLAM per se, and not on more on fuller range of hybrid and theoretical possibilities that are open to you, given we have this huge decision making literature.

We thank the reviewer for this suggestion. We fully agree and have therefore developed and compared a set of models that span the space between satisficing and gaze-driven evidence accumulation. For a detailed overview of these models and the results of the model comparison, see our response to your remark #1.

– The behavioral analysis is interesting, and to me under-played. Looking into individual subjects differences, and spending the time to highlight consistencies with the averages would be worthwhile. This is new task and I don't leave the paper understanding its basic results as well I would like.

We are thankful for this suggestion and have extended our empirical analysis of the data on two accounts:

First, we now provide an analysis of the association of the different measures of choice behaviour (see Figure 4F of our revised manuscript; namely, mean RT, fraction of looked-at items in a trial, probability of choosing the highest-rated seen item, and gaze influence on choice probability). This analysis indicated that mean RT increases with the fraction of look-at items in a trial. It further provided additional support for the assumption of gaze-driven evidence accumulation. Specifically, we found that the probability of choosing the highest-rated last seen item decreases with an increasing influence of gaze on choice probability. A strong influence of gaze on choice thereby shifts choice probabilities away from the liking values towards the distribution of gaze (in line with the findings reported in Thomas, Molter, Krajbich, Heekeren and Mohr, 2019). Further, we found that the probability of choosing the last seen item in a trial increases with an increasing influence of gaze on choice; the assumption of gaze-driven evidence accumulation predicts a bias towards choosing the last seen item as evidence is generally accumulated at the highest rate for the momentarily looked-at item.

Second, we tested the association between the different measures of choice behaviour and a set of measures describing individuals’ visual search (Figure 4—figure supplement 2). To quantify individuals’ visual search behaviour, we ran a regression for each subject of cumulative gaze (defined as the fraction of trial time that the subject looked at an item; scaled 0 – 100 %) onto the four item attributes (row, column, size, and rating) and set size. These coefficients thereby indicate the influence of these four item attributes on an individuals’ distribution of gaze. This analysis indicated that individuals with a stronger influence of rating on gaze allocation generally looked at fewer items, were more likely to choose the highest-rated seen item, and were more likely to choose the last-seen item. Individuals with a stronger influence of item size on gaze allocation generally looked at fewer items, exhibited shorter RTs, and were less likely to choose the last seen item. Lastly, individuals with a stronger influence of column number (horizontal location) on gaze allocation generally exhibited longer RTs.

Resulting changes to the manuscript:

“To further probe the assumption of gaze-driven evidence accumulation, we performed three tests: According to the framework of gaze-driven evidence accumulation, subjects who exhibit a stronger association between gaze and choice should generally exhibit a lower probability of choosing the highestrated item from a choice set (for a detailed discussion on this finding, see Thomas et al., 2019). […] In line with this prediction, subjects with a stronger relation between gaze and choice were generally also more likely to choose the item that they looked at last (β = 1.1%, 94% HDI = [0.9, 1.3] per percentage increase in gaze influence; the mixed-effects regression included a random slope and intercept for each set size).”

“To better understand the relationship between visual search and choice behavior, we also studied the association of the influence of an item’s size, rating, and position on gaze allocation with the metrics of choice behavior reported in Figure 4 (namely, mean RT, fraction of looked-at items, probability of choosing the highest-rated seen item, and gaze influence on choice) (see Figure 4—figure supplement 2). […] Lastly, subjects with a stronger influence of column number (horizontal location) on gaze allocation generally exhibited longer RTs (β = 3.9 s, 94% HDI = [0.1, 7.8] per unit increase in β_gaze_(column); Figure 4—figure supplement 2 J). We did not find any other statistically meaningful associations between visual search and choice metrics (see Figure 4—figure supplement 2).”

[Editors' note: further revisions were suggested prior to acceptance, as described below.]

Revisions:1) New model fitsReviewer #1 has one final concern with the manuscript, resulting from the new model fits; it does not become entirely clear how the reader should think about gaze-dependent choice behavior in this task. The PSM+ seems to be the best model overall, but cannot account for the specific relationship between choice and gaze; and the relative and independent evidence accumulation models each have their advantages depending on the set size. After reading, this leaves the reader wondering how tey should best think about the mechanisms of choice behavior in this task.Now, the data are of course the data. We do not expect the authors to make overblown claims about one winning model, if this is not supported by the data. But a bit more interpretation and context throughout, as well as in the Discussion, would help guide the reader; how do the authors think choice behavior most likely comes about? If they can't distinguish between some models (depending on the metric used), why does this conflict arise? Are there future modelling efforts or experiments that you can propose that would clarify this issue?

We thank the reviewer for raising these very important questions. We have expanded our analyses of the individual model comparisons to provide a better answer to these questions:

At first sight, the PSM+ and IAM+ seem to be winning the majority of individual WAIC comparisons over the GLAM+ (Figure 5A-D). However, when comparing the overall distributions of WAIC values within each set size (by the use of two-sided Mann–Whitney U tests with a Bonferroni adjusted α level of 0.0042 per test (.05/12), we find that these wins are only by a small margin (Figure 5E-H)). Specifically, the WAIC distributions of the PSM+ and GLAM+ were not meaningfully different from one another in any set size (U = 1287, p = 0.54; U = 1444, p = 0.08; U = 1460, p = 0.07; U = 1469, p = 0.06 for 9, 16, 25, and 36 items), while the PSM+ was meaningfully better than the IAM+ for 16 items (U = 1705, p = 0.0003), but not for 9 (U = 1592, p = 0.005), 25 (U = 1573, p = 0.008) or 36 (U = 1508, p = 0.029) items. The GLAM+ was not meaningfully better than the IAM+ in any set size (U = 1518, p = 0.02; U = 1507, p = 0.03; U = 1388, p = 0.18; U = 1289, p = 0.53 for 9, 16, 25, and 36 items). We also find that the difference in WAIC between the PSM+ and IAM+ as well as the PSM+ and GLAM+ increased with set size (β = 0.55, 94% HDI = [0.16, 0.93] per item for the IAM+ and β = 0.86, 94% HDI = [0.62, 1.12] per item for the GLAM+), indicating that the PSM+ provides an increasingly better fit to the choice and RT data as the set size increases. Notably, the corresponding fixed-effects intercept estimate was larger than 0 for the IAM+ (19.77, 94% HDI = [13.17, 26.55]) but not for the GLAM+ (-0.7, 94% HDI = [-5.3, 3.9]), suggesting that the PSM+ also provides a meaningfully better fit for choice behaviour in smaller sets than the IAM+, but not than the GLAM+. Similarly, the WAIC-difference between the GLAM+ and IAM+ did not increase with set size (β = -0.30, 94% HDI = [-0.73, 0.15] per item), while the fixed-effects intercept estimate was meaningfully greater than 0 (20.52, 94% HDI = [14.95, 26.06]), indicating that the GLAM+ provides a meaningfully better fit than the IAM+ for smaller sets.

Further, when analyzing the choice and RT predictions of the three models, we find that the PSM+ and GLAM+ perform similarly well in capturing individuals’ choices and RTs and both outperform the IAM+ (Figure 6A-F). In contrast, the GLAM+ clearly outperforms the other two models in capturing individuals’ association of gaze and choice (Figure 6G-I), which is not accounted for in the individual WAIC comparisons. We believe that this highlights a weakness in evaluating model fit purely based on choices and RTs, as these do not account for other aspects of the decision process (such as eye movements). We have revised our manuscript throughout to better highlight this perspective.

Resulting changes to the manuscript:

Abstract

“A gaze-driven, probabilistic version of satisficing generally provides slightly better fits to choices and response times, while the gaze-driven evidence accumulation and comparison model provides the best overall account of the data when also considering the empirical relation between gaze allocation and choice.”

Introduction

“The probabilistic satisficing model performs slightly better than the other models at capturing individuals’ choices and RTs, while the relative accumulator model provides the best overall account of the data after considering the observed positive relation between gaze allocation and choice.”

Results:

“According to WAIC, the choices and RTs of the vast majority of subjects were best captured by the model variants with an active account of gaze (82% (40/49), 94% (46/49), 90% (44/49), and 86% (42/49) for 9, 16, 25, and 36 items respectively; Figure 5A-D). […] Notably, the corresponding fixed effects intercept estimate was larger than 0 for the IAM+ (19.77, 94% HDI = [13.17, 26.55]) but not for the GLAM+ (-0.7, 94% HDI = [-5.3, 3.9]), suggesting that the PSM+ also provides a meaningfully better fit for choice behaviour in smaller sets than the IAM+, but not than the GLAM+. Similarly, the WAIC-difference between the GLAM+ and IAM+ did not increase with set size (β = -0.30, 94% HDI = [-0.73, 0.15] per item), while the fixed effects intercept estimate was meaningfully greater than 0 (20.52, 94% HDI = [14.95, 26.06]), indicating that the GLAM+ provides a meaningfully better fit than the IAM+ for smaller sets.”

Discussion:

“After incorporating active effects of gaze into a probabilistic version of satisficing, it explained the data well, slightly outperforming the evidence accumulation models in fitting choice and RT data. Yet, the relative accumulation model provided by far the best fit to the observed association between gaze allocation and choice behaviour, which was not explicitly accounted for in the likelihood-based model comparison, thus demonstrating that it provides the most comprehensive account of behaviour in MAFC.”